# Microvalves for Applications in Centrifugal Microfluidics

**DOI:** 10.3390/s22228955

**Published:** 2022-11-18

**Authors:** Snehan Peshin, Marc Madou, Lawrence Kulinsky

**Affiliations:** 1Department of Mechanical and Aerospace Engineering, University of California, Irvine, CA 92697, USA; 2School of Engineering and Science, Tecnológico de Monterrey, Monterrey 64849, Mexico

**Keywords:** centrifugal microfluidics, microfluidic valving, point-of-care diagnostics, Lab-on-CD

## Abstract

Centrifugal microfluidic platforms (CDs) have opened new possibilities for inexpensive point-of-care (POC) diagnostics. They are now widely used in applications requiring polymerase chain reaction steps, blood plasma separation, serial dilutions, and many other diagnostic processes. CD microfluidic devices allow a variety of complex processes to transfer onto the small disc platform that previously were carried out by individual expensive laboratory equipment requiring trained personnel. The portability, ease of operation, integration, and robustness of the CD fluidic platforms requires simple, reliable, and scalable designs to control the flow of fluids. Valves play a vital role in opening/closing of microfluidic channels to enable a precise control of the flow of fluids on a centrifugal platform. Valving systems are also critical in isolating chambers from the rest of a fluidic network at required times, in effectively directing the reagents to the target location, in serial dilutions, and in integration of multiple other processes on a single CD. In this paper, we review the various available fluidic valving systems, discuss their working principles, and evaluate their compatibility with CD fluidic platforms. We categorize the presented valving systems into either “active”, “passive”, or “hybrid”—based on their actuation mechanism that can be mechanical, thermal, hydrophobic/hydrophilic, solubility-based, phase-change, and others. Important topics such as their actuation mechanism, governing physics, variability of performance, necessary disc spin rate for valve actuation, valve response time, and other parameters are discussed. The applicability of some types of valves for specialized functions such as reagent storage, flow control, and other applications is summarized.

## 1. Introduction

Centrifugal microfluidic devices (CDs) were successfully implemented for point-of-care (POC) diagnostic platforms. POC systems are platforms that can be implemented at the bedside of the patient or in a medical office, away from the centralized laboratory. They help in reducing the turnaround time for the patients and doctors to receive the test results, contributing to improved patient outcomes and reducing treatment cost (by accelerating the time to diagnosis and by eliminating the need for repeat patient visits if the tests can be performed during the initial doctor’s visit). These POC systems are robust, portable, inexpensive, and easy to operate, even by moderately trained personnel.

Since the first introduction of the centrifugal platform in 1972 by Anderson et al. [1], CD technology has come a long way and is now being used for a wide range of processes, including immunoassays, environmental monitoring, detection of analytes, serial dilutions, and numerous other applications. The disc-based platform LabCD was introduced by Madou and Kellogg [2] in the late 1990s, and since that time, there has been a continuous stream of publications from various academic groups and industrial teams on adapting centrifugal microfluidic platforms to various diagnostic purposes—from the enzymatic assay analyzers [3] and specialized modules for disc-based immunoassay microarrays [4], to the characterization of pollutants in environmental samples [5], detection of antibiotic resistance on a CD platform [6], and a development of a portable CD system for biological and analytical testing [7].

Active developments and enhancements in CD technology that took place in the last 20 years were supported by the advances in fidelity and reliability of microfluidic valving—a critical fluidic operation on a spinning disc. Valves control the flow of fluids, switch channel paths on/off, isolate specific chambers, and open chambers for the serial release of reagents in a controlled sequential manner. Initially, only so-called “passive valves” such as capillary valves [8], hydrophobic valves [9], and siphon valves [10] were used. These passive valves did not require any peripheral equipment for actuation, they were easy to fabricate and simple to operate and mostly were based on the opposing action of capillary forces (controlled by the material of the disc and the geometry of the microfluidic channels) and of the centrifugal forces (controlled by the angular velocity of the disc and specific position and geometry of the microfluidic channel) on the CD. Therefore, the action of passive valves depends on such parameters as spin speed [11], the geometry of channels and chambers, the location of the vents on the discs (that allow the pressure in various microfluidic chambers to be equal to the ambient pressure), the type of the native material of the disc and the various coatings that are used to modify the wetting angle of the liquid on the surface [12]. Table 1 presents a list of passive valves classified into categories according to their controlling parameters, such as spin speed, vent-hole geometry, suction, channel divergence, concentration gradient, presence of siphon channels, and the inclusion of dissolvable films.

Gradually, as CD devices started to be utilized in commercial applications beyond academia, researchers and engineers have realized that manufacturing imperfections and minor differences in the bill of materials and materials’ shelf life affect reliability and repeatability of operations of passive valves. To mitigate issues with robust operations of passive valves, active valves were introduced. Active valves use some external subsystems to trigger and actuate opening or closing the fluidic channel. The typical actuation mechanisms for active valves, such as laser actuation [13]; magnetic actuation [14]; diaphragm-based actuation [15]; electrical [16], thermal [17], mechanical, and pH-controlled actuations; and other mechanisms of valve actuations, are also summarized in Table 1.

Finally, there is still yet another type of valves, so-called “hybrid valves”, that utilize elements of both active and passive valves. Examples of hybrid valves include the ferrowax capillary flow-driven valve [13], microheater activated valve [17], and similarly actuated valves. Such valves are based on capillary forces native to the microfluidic system of the platform, but the triggering of the actuation of such a valve requires external equipment (hot plate, laser, etc.) that converts the solid wax plug into a liquid form. Once the phase transformation changes the plug into a liquid form, the plug can move, and the fluidic channel is opened.

Active and hybrid valves are more reliable than passive valves and typically have lower response times. In the sections below, various types and categories of valves are discussed in detail, including valves’ operational parameters such as response times, fabrication routes, and manufacturing cost. This information offers readers greater insight into choosing the appropriate type of valve for their specific application. Table 1 presents the maximum resistance revolutions per minute (RPM) and response time for each of the valves discussed in this article. In Table 2 we present the various applications for each type of valve.

The present critical review focuses on the fabrication and application of microvalves on centrifugal microfluidic platforms. Our work follows on and updates more general reviews of microvalves on lab-on-chip platforms [18,19,20], as well as includes a discussion of the virtual prototyping of many types of microvalves implemented on centrifugal microfluidic platforms [21].
sensors-22-08955-t001_Table 1Table 1Classification of valves and their compatibility with CD fluidics, costs, and response times.Valve TypeControlNameMaximum Resistance (RPM)Response Time Fabrication MethodsApplicationsPassiveDisc Spin speed or directionPouch-based micro-liquid dispensing [11]5300 2 sTube sealing and drillingReagent storage; Flow switching

Stickpacks [22,23,24]3900 5 sUltrasonic weldingReagent storage; Large sample volume processing

Coriolis force microvalve [25,26]4800 1 sLaminated object manufacturingFlow switching; Serial dilution

Air plug-based resistance switch valve [27]4800 1 sLaminated object manufacturingFlow switching; Serial dilution

Euler force microvalve [28]--Laminated object manufacturingBlood plasma separation; Metering operation
Pressure-BasedVacuum compression valve [29]1100 1 sWax applicationFlow control; Flow switching 

Passive liquid valve [30]11001 sLaminated object manufacturingFlow control; Flow switching

Water clock microvalve [31]1100 variableLaminated object manufacturingSerial dilution; Large sample volume processing
Capillary-BasedDiverging channel-based burst capillary microvalve [8,32,33,34]500 1 sLaminated object manufacturingFlow switching; Metering

Suction-enhanced burst capillary microvalve [35]300 1 sLaminated object manufacturingFlow control; Sample volume processing
ConcentrationConcentration valve [36]-45 minConcentration sensitive materialFlow control; Flow switching 
Dissolvable filmsDissolvable film microvalve [37]430010 sAssembly of filmsFlow switching; Single use
Hydrophobic patch-basedHydrophobic microvalve [9,38,39]3001 sSurface treatmentFlow switching; Metering 

Aliquot microvalve [40]1500 1 sSurface treatmentMetering; Small sample volume processing
SiphonSiphon microvalve [10,41,42,43]60002 sLaminated object manufacturingSample volume isolation; Metering2ActiveLaser actuationLaserPacks [44]6500 2 sThermoformingReagent storage; Large sample volume processing

Laser printer valve [45,46]6500 1 sAssembly of filmsFlow switching; Single use

Foam-based [47]1000 14 minApplication of foamFlow switching; Flow control
Magnetic actuationReversible magnetic wax valve [48]6500 6 sApplication of ferrowaxFlow switching; Multi-use
Mechanical actuationSoft diaphragm valve [15]6000 2 sSoft lithographyFlow control; Flow switching

Layer-based diaphragm valve [49]6000 1 msSoft lithographyFlow control; Flow switching

Elastomeric diaphragm valve [50]6000 1.5 sSoft lithographyFlow control; Flow switching

Magnetic ball valve [51]6000 1 sAssembly of magnetsFlow control; Flow switching

Pneumatic soft valve [52]6000 3 sSoft lithographyFlow control; Flow switching

Soft-lithography-based pinch valves [53,54]6000 3 sSoft lithographyFlow control; Flow switching
Environmental-sensitive valvespH-controlled hydrogel valve [55]300 10 sDirect photopatterning of a liquid phaseFlow control; Flow switching

Thermally soluble polymer valve [56]6000 4 sPhotoinitiated polymerizationMulti-use valve; Flow switching

Gel-based valve [57]-5 msFree-radical copolymerizationMulti-use valve; Flow switching 

Temperature memory gel valve [58]-20 minInterpenetration of one gel network with anotherMulti-use valve; Flow switching

Thermally actuated membrane valve [59]60009 sAssembly of heating element and soft lithographyFlow control; Flow switching
Phase-change-based valveSolid–liquid phase change valve [60,61]6000 10 sUse of Peltier coupleFlow control; Flow switching

Wax-based valve [62]400 5 sApplication of wax and laserFlow switching; Multi-use

Vacuum wax valve [63]400 5 sApplication of wax and vacuumFlow switching; Multi-use
Electrically activatedElectrically controlled valve [16]600060 sApplication of electrical elementFlow control; Multi-use
Pressure-drivenAir-pressure-based check valves [64]480 2 sApplication of vent seatsFlow switching; Multi-use

Xurography-based valve [12]4300 10 sDissolvable film applicationFlow switching; Single use

Hydrophobic laser printer lithography valves [65]30010 sLaser printer lithographyFlow switching; Easy patterning 3HybridCapillary-drivenCapillary driven ferrowax valve [13,48]650015 sApplication of ferrowax and laserMulti-use; Flow switching
Microheater actuationLIFM [14,17,66]4005 sApplication of heating element and a Peltier coupleFlow switching; Single use
sensors-22-08955-t002_Table 2Table 2Classification of valves based on their fit for specialized applications.Valve Type/ApplicationsReagent StorageFlow ControlFlow StopMultiple UseLow LeakageSample VolumeSection 2.1.1GoodModerateGoodGoodModeratePoorSection 2.1.2GoodPoorGoodPoorGoodGoodSection 2.1.3ModerateModeratePoorGoodModerateGoodSection 2.1.4PoorPoorPoorGoodPoorGoodSection 2.1.5PoorPoorModerateGoodGoodGoodSection 2.1.6ModerateModerateGoodGoodModerateGoodSection 2.1.7PoorPoorGoodGoodGoodGoodSection 2.1.8PoorPoorGoodGoodGoodGoodSection 2.1.9GoodPoorGoodGoodGoodGoodSection 2.1.10GoodPoorGoodGoodGoodGoodSection 2.1.11PoorGoodGoodGoodGoodGoodSection 2.1.12PoorPoorGoodGoodGoodModerateSection 2.1.13PoorGoodGoodGoodModerateGoodSection 2.1.14ModeratePoorGoodPoorGoodGoodSection 2.1.15PoorGoodModerateGoodModerateGoodSection 2.2.1GoodPoorGoodPoorGoodGoodSection 2.2.2GoodPoorGoodPoorGoodGoodSection 2.2.3GoodPoorGoodPoorGoodGoodSection 2.2.4GoodPoorGoodGoodGoodGoodSection 2.2.5GoodPoorGoodGoodGoodGoodSection 2.2.6ModerateGoodGoodGoodModerateGoodSection 2.2.7ModerateGoodGoodGoodModerateGoodSection 2.2.8ModerateGoodGoodGoodModerateGoodSection 2.2.9ModeratePoorGoodGoodModerateGoodSection 2.2.10ModerateModerateGoodGoodGoodGoodSection 2.2.11ModerateModerateGoodGoodGoodGoodSection 2.2.12PoorGoodModerateGoodModeratePoorSection 2.2.13ModeratePoorModerateGoodModerateModerateSection 2.2.14PoorGoodModerateGoodModerateModerateSection 2.2.15PoorGoodGoodGoodModerateModerateSection 2.2.16GoodGoodGoodModerateGoodGoodSection 2.2.17ModeratePoorGoodGoodGoodGoodSection 2.2.18GoodPoorGoodGoodGoodGoodSection 2.2.19GoodPoorGoodGoodGoodGoodSection 2.2.20GoodModerateGoodGoodGoodGoodSection 2.2.21ModeratePoorGoodPoorGoodGoodSection 2.2.22PoorPoorGoodGoodGoodGoodSection 2.3.1 and Section 2.3.2GoodPoorGoodGoodGoodGood


## 2. Detailed Description of Microvalves

### 2.1. Passive Microvalves

Passive microvalves are those microvalves that do not require additional peripheral devices or systems for their operation. The actuation of passive microvalves on a CD platform is based on parameters that include spin speed, spin direction, chamber and channel dimensions, etc. Passive microvalves, as shown in Table 1, may be categorized based on their actuation method and parameters such as response time and maximum resistance to the fluidic flow in RPM of CD. The most frequently utilized control parameters are the angular velocity of the rotating disc [11,22,23,24], as well as the spin direction [27], and these parameters define the governing centrifugal force as well as the Euler and Coriolis forces [25,26,28]. These conditions may be easily modified to fine-tune the timing of the dispensing of fluids from various chambers. Jens Ducree and colleagues recently published a review [21] that lists common models of valving schemes at the heart of “Lab-on-a-Disc” (LoaD) platforms to enable virtual prototyping and algorithmic design optimization of centrifugal microfluidic platforms. While the operation of the CDs to actuate the passive microvalves is relatively straightforward, it is the precision of fabrication and the robust design of the fluidic elements on the disc that warrant the reliable operation of the passive valves, and these aspects of design and fabrication are discussed in the subsections below.

#### 2.1.1. Pouch-based Micro-Liquid Dispensing [11]

The pouch-based micro-liquid dispensing [11] valve is a pouch-shaped tube (with a hole covered by a ribbon of stretchable membrane) that is used for storage and controlled release of liquids on a centrifugal platform, as shown in Figure 1a. Due to the increase in the pressure of the fluid in the tube onto a membrane (caused by the centrifugal force present on the spinning disc), the liquid is dispensed through the hole as the membrane is stretched.

The physics of the dispensation in the pouch is described by derivation of the physical equation by assuming some simplifications. We assume pressure *p* at *z* = *h* is above a critical pressure p_o_, which depends on the membrane properties and assumes the tube is rigid. We also assume a small cross-sectional area of the tube. Finally, we neglect the friction between the tube and the membrane. By integrating the centrifugal force from base of the tube to height *h*, we obtain the pressure from rotation at spin speed ω as:(1)$$p\left(z\right)=\frac{1}{2}\rho \omega^{2}\left[{\left(H-h\right)}^{2}-{\left(H-z\right)}^{2}\right]$$
here, *z* is the level of the fluid in the tube from the base of the tube, ρ is the density of the fluid, ω is the spin rate of the CD, H is the radial distance of the base of the tube, *h* is the location of the hole from the base of the tube.

The pouch is fabricated using an impermeable tube that is cut into strips of several cm long (length and the diameter of the tube can vary). The tube is sealed at both ends by thermal sealing (heated above the melting point and then cooled to fuse the ends). Afterwards, a hole is drilled in the tube; the location and height of the hole is determined based on the level of the fluid that must remain in the tube after the fluid is dispensed. A neoprene band with an inner diameter less than the outer diameter of the tube is fitted over the tube. As the disc spins, the centrifugal force exerted by the fluid in the tube causes the increase in the pressure on the elastic band until, at the point above some critical angular velocity, the band stretches sufficiently to allow the fluid to exit from the pouch. The geometry and location of the tube, as well as the thickness and the inner diameter of the ribbon, determine the critical angular velocity of the spinning disc at which the fluid is released from the tube. It can be suspected that variability in the quality of elastic bands and change in the stiffness/elasticity of the bands with time (and thus, influence of shelf time on performance of the device) will negatively affect pouch-to-pouch variability and can serve as an impediment to translating this technology to commercial production.

The quick response time (several seconds of spinning at the critical angular velocity) and the low cost of the pouch makes this storage and valving combined system a promising option for commercial CD platforms.

#### 2.1.2. Stickpacks [22,23,24]

The use of stickpacks on the centrifugal platform was introduced by the Zengerle group in 2013 [19]. Structurally, stickpacks are tubular-shaped composite foil pouches rectangular packs comprised of a one-side-folded sheet of 12 µm polyethylene terephthalate (PET), 9-micron aluminum, and a layer of 70 µm polyethylene (PE). The PE is sealed on the shorter (transverse) edges, creating a temporary seal, while the longer (longitudinal) edge is secured by a permanent seal, as shown in Figure 1b. This structure controls the dispensing of fluid through the shorter seal at the bottom side, as actuated by the centrifugal force acting on the liquid-filled cavity.

The author inserted these packs on a CD and calculated the burst pressure using the following equation:(2)$$p_{liquid}=\frac{1}{2}\rho {\left(2\pi f\right)}^{2}\left(r_{2}{}^{2}-r_{1}{}^{2}\right)$$
where ρ is the density of the fluid stored in the stickpacks, *f* is the spinning frequency, and $r_{1}$ and $r_{2}$ are the innermost and outermost radius of the liquid column, respectively.

Initially, the Zengerle group proposed the use of slow ultrasonic welding to seal stickpacks, where a thicker anvil presses the longitudinal seal and a smaller anvil presses the transverse seal (ECO35 welder, sonotrode type Ti, RSMS 33/21/10, L1/2 by Sonotronic, Karlsbad, Germany). Further development by that group, described in a 2015 paper [18] introduced a less expensive and more scalable process of thermal sealing (SBL50, Merz Verpackungsmaschinen GmbH, Lich, Germany). Thermal sealing involves heating the sealing region with the simultaneous application of high pressure by a vibrating anvil. In the case of thermal sealing, the strength of the seal is a function of temperature, as well as the magnitude and frequency of the applied pressure, and the duration of the thermal sealing process (150 kPa, 100 °C). The fabrication parameters and the dimensions of the stickpacks were optimized as the thermal sealing substituted the slow ultrasonic sealing. For example, the volume range of stickpacks increased from 80–500 μL to 50–1200 μL; the dimensions of the scalable stickpacks were either 9 mm or 15 mm in width and 10 to 100 mm in length for thermal sealing compared to 5–10 mm width and 10–20 mm length for the ultrasonic welding; and a minimum cycle time of 1.5 s using the new thermal sealing process as compared to more than 5 s for the ultrasonic welding.

The burst frequency in revolutions per minute (RPM) depends on the radial location of stickpacks on the CD, the volume of fluid in the stickpack, and the sealing parameters of the transverse edge (i.e., temperature, pressure, and duration of thermal sealing). The stickpacks dispense the liquid by spinning the CD when an angular velocity is higher than the stated stickpack’s burst frequency.

It is reasonable to assume that there is a certain variation in the critical rpm required for bursting the packs in both, pack-to-pack difference as well as change in critical burst rpm depending on the storage time and storage conditions of the packs. It is also likely that thermal sealing would limit the type of reagents that can be stored in the packs.

The stickpacks were designed specifically to be actuated by centrifugal force and, thus, they are compatible with CD fluidics. 

#### 2.1.3. Coriolis Force Microvalve [25,26]

The Coriolis force microvalve was introduced by the Zengerle group in 2005 [25]. The functionality of this valve depends on the direction of spin where the fluid switches between two channels that are bifurcated at the junction, leading to two reservoirs or fluidic networks, as is illustrated in Figure 2a. If the valve spins in the clockwise direction, the fluid flows into the right chamber, and in case of the counterclockwise direction, it enters the left chamber. The author characterized the action of the valve as a function of the geometry of the bifurcated fluidic networks.

The Coriolis component of the force can be related to the centrifugal force by using the derived equation as: (3)$$\frac{\left|{\underset{f}{\to }}_{coriolis}\right|}{\left|{\underset{f}{\to }}_{\theta }\right|}=\frac{\rho^{2}\Delta x^{3}l\omega^{3}\bar{r}}{16\sigma \eta cos\Theta }$$
where the ratio of force density is shown to be proportional to $\omega^{3}$. Here, ρ is the density of the fluid, delta x is the dimension of the channel, l is the length of the channel, ω is the angular velocity, $\bar{r}$ is the radial vector, σ is the surface tension, η is the viscosity, and Θ is the contact angle. Furthermore, 75 rad/s is the equilibrium angular velocity above which the Coriolis forces dominate and help in switching the fluid and actuate the valve.

The Coriolis valve’s drawback relates to some uncertainty in the amount of fluid that would go into one or another of the bifurcating channels during the process of dynamic switching of the direction of rotation of the disc. The Coriolis valve is not designed to stop the flow but rather to route the fluid to the appropriate branch of the fluidic network.

#### 2.1.4. Air Plug Valve [27]

The air plug-based resistance switch valve was first introduced by Kim et al. in their 2008 paper [27]. The valve utilizes both the geometry of the fluidic network and the air trapped within the channel (controlled by the presence or absence of the vent holes on a disc). As shown in Figure 2b, the channels are designed asymmetrically—the wider channel leads to the right chamber and the smaller tilted channel leads to the left chamber. When the fluid level in the right chamber reaches the height of the valving channel entrance or higher as required for the process, the CD is stopped to create an air plug trapped in the channel that prevents the intake of additional fluid into that chamber, and once the disc spinning is resumed, the fluid starts flowing into the left chamber through the narrow channel.

This type of microvalve does not need external materials for fabrication and, thus, the cost and complexity of the manufacturing process are reduced. A major disadvantage for this design is that it cannot be used for more than two on/off switches, and this type of valve requires a large space on a CD.

As this valve is based on a centrifugal pumping mechanism, it is compatible with the CD platform. Because this type of valve does not require additional manufacturing steps, the cost of the valve is low, but stopping and restarting the disc and reliance on the specific amount of trapped air to enable the switching would require stringent quality control and repeatability of both, disc manufacturing and spin control to ensure a reliable and robust operation for such a gating valve. The response time for these valves is only 1 or 2 s after the disc starts spinning.

#### 2.1.5. Siphon Microvalve [10,41,42,43]

The siphon microvalve is widely used for many applications on the CD platform [10,41,42,43], including blood plasma separation and serial dilutions. The physical principle of the action of the siphon valve is that small hydrophilic siphon channel (typically having a “U”-shape) would be wetted by a liquid from the connected chamber if the disc would not be spinning. However, the spinning disc produces centrifugal force that pushes the liquid radially downward, not allowing the fluid to enter the siphon channel and to reach the top crest of that siphon channel. When the disc is stopped, the capillary force then carries the fluid through the siphon channel, across the top of the channel, and then primes the siphon, initiating the fluid flow between the inlet chamber and the outlet chamber that are connected by the siphon.

To demonstrate the action of the siphon valve, it is instructive to examine (see Figure 3) the blood plasma separation as a venue for application of the siphon valve. In the blood plasma separation, the denser red blood cells are sedimented at the bottom of the separation chamber, and the watery plasma is flown to the top of the separation chamber. When the disc rotates and the red blood cells are separated from plasma, the fluid cannot rise over the crest of the siphon channel due to the centrifugal force. When the plasma is separated from the red blood cells in the separation chamber, the disc is stopped and the plasma is transferred via the siphon (extraction) channel under the influence of the capillary force.

One of the major issues related to reliability of the siphon microvalves is dependance of the capillary force advancing or retarding the flow to the surface condition of microchannels. Variation of the specified operating conditions (including the elevated temperature during disc operation) or the change in the contact angle of the surface of microchannels during the time between disc’s production and use can lead to valve failure.

The siphon microvalve is compatible with centrifugal microfluidics. It is simple to design and fabricate, but it does take extra space on the disc for the placement of the siphon extraction channel.

#### 2.1.6. Euler Force Microvalve [28]

The Euler force microvalve, first introduced by Deng et al. in 2014 [28], is a subtype of siphon microvalve that is actuated by the Euler force. The Euler force is the force acting in the transverse direction (normal to the radial direction) when the disc undergoes angular acceleration. The Euler force provides an extra push for the fluid in the siphon (see Figure 4) to rise over the top of the crest of the siphon and, thus, contribute to initiation of the flow from the metering chamber through the siphon into the extraction chamber. The same Euler force valve design was used for the blood plasma separation discussed in the same publication. The centrifugal force on the spinning disc facilitates fast sedimentation and separation of the red blood cells that are denser than the surrounding plasma. Blood plasma from the metering chamber is then transferred to the extracting chamber.

The actuation of this microvalve uses the initial dominance of Euler force, and then, as the ω increases, the dominance of the centrifugal force initiates the dispensation of the fluid in the radial direction. As shown below, the ratio of Euler force to centrifugal force is: (4)$$\frac{\left|-\rho \alpha \times r\right|\cos\left(\frac{\pi }{2}-\beta \right)}{\left|\rho \omega \times r\times \omega \right|}=\frac{\left|\alpha \right|sin\beta }{{\left|\omega \right|}^{2}}\gg 1\;or\ll 1$$
where ρ is the density, α is the angular acceleration equal to dω/dt, r is the rotating radius vector, u is the fluid velocity, β is the angle of the channel, and ω is the angular velocity of the disc.

The major advantage of this microvalve is that it needs a siphon design only and can be actuated by varying the spin rate of the CD. A major disadvantage is that for the siphon to prime, we need very precise CD spin rate manipulations that, if not performed correctly, can lead to the failure of the microvalve. Variations in the hydrophobicity of the material of the disc and sidewalls of the siphon channel will also affect the reliability of this valve.

Despite its complex structure, the costs associated with the manufacturing of this disc remain low, attributed mainly to fabrication of the disc itself.

#### 2.1.7. Diverging Channel-Based Burst Capillary Microvalve [8,32,33,34]

In the capillary burst microvalve (CBV), one of the earliest passive valves implemented on CDs, the valving principle is based on the opposition of the surface tension forces, holding the liquid meniscus at the exit of the fluidic channel, and the centrifugal forces. In a generalized case, a capillary burst valve utilizes a narrow channel that diverges at a wedge angle (β) into a wider chamber/channel, as shown in Figure 5a. The movement of fluid through a CBV commences at the point at which the spin speed of the CDs surpasses the burst frequency of the CBV. The wedge angle parameter β can be varied to control and optimize the burst frequency of the CBV. As β increases, the burst frequency of the microvalve also increases. Other tunable parameters include the channel aspect ratio (here, the channel aspect ratio is the ratio of the height h of the channel to the width w of the channel) and the burst frequency increases with the increase in the channel aspect ratio. In the Figure 5b we see the bird’s eye view of this valve where h is the height of the channel and w is the width of the channel.

We also discuss here the theory of the burst valve and its condition for pressure in a rectangular channel as shown in Figure 5. Here the contact angle with the sidewalls is θ_s_ and with the top and bottom walls is θ_v_. The Youngs Laplace equation yields:(5)$$P_{A}-P_{o}=-2\sigma \left(\frac{cos\theta^{*}{}_{s}}{b}+\frac{cos\theta_{v}}{h}\right)$$
where *w* and *h* are the width and height of the channel, respectively. *b* denotes the width of the diverging channel. Until the pressure difference reaches the threshold valve, the valve does not burst. For a narrowing channel we derive the following relationship to obtain the threshold angular velocity for the bursting valve: (6)$$\omega_{b}={\left[4\sigma \frac{\frac{cos\theta_{R}-cos\theta_{v}}{h}+\left(\frac{cos\theta_{R}}{c}-\frac{cos\theta^{*}{}_{s}}{w}\;\right)}{\rho \left(r_{2}{}^{2}-r^{2}{}_{1}\right)}\right]}^{1/2}$$

Here, $\omega_{b}$ is the burst frequency for the valve, ρ is the density of the fluid, r1 and r2 are the receding and advancing fronts of the slug, θ is the contact angle (R for the critical receding contact angle), and c is the wider channel width.

One of the major advantages of this microvalve is the small space requirement on the CD design space. The major disadvantage (other than the typical dependance of the passive valves on fabrication variations and on the change in materials’ properties with time) is that unless extremely small channels are used (posing fabrication challenges), the capillary valves would burst at a relatively low angular velocity of the disc (typically below 1000 rpm). For the robust operation of various multi-step assays on CD platforms, the sequence of steps where the fluid is dispensed from reservoirs should utilize some safety margin for the burst frequency of each valve; therefore, if the top angular velocity to burst valves is low, it presents a significant challenge for designing multi-step assays with CBVs, and we might require separate CDs (used sequentially) to run multi-step processes.

Because manufacturing imperfections significantly influence the burst frequency of CBV, there is a considerable variation of burst frequencies among valves of this type. Additionally, surface properties of the microchannels change with time, and this change also affects the burst frequency of the CBV valves. Finally, because it is difficult to produce reliable channels with widths of less than 200 microns via the injection molding technique, there is a limit on how high the burst frequency can rise for CBV valves.

CBVs are compatible with centrifugal platforms, and there is an insignificant fabrication cost involving the molding and designing of the CDs with these valves.

#### 2.1.8. Hydrophobic Microvalve [9,38,39]

Hydrophobic microvalves [9,38,39] use patches of hydrophobic materials for the channel wall (or multiple sides of the microfluidic channel) to constrain the passage of the liquid in the channel due to capillary forces. When implemented on CDs, the hydrophobic valve will burst when the angular velocity exceeds the burst frequency of the valve. The geometry of the hydrophobic patch (cross-sectional geometry of the channel, the number of hydrophilic and hydrophobic sidewall of the channel, the length of the hydrophobic patch), as well as the contact angles of the hydrophobic and hydrophilic materials composing the valve, affect the burst frequency of such a valve. 

Similar to the CBVs, the major advantages of this microvalve are the compact design, the simple and robust action of the hydrophobic patch, and its affordability. Major disadvantages are that this microvalve cannot be used for non-aqueous samples/analytes and that additional fabrication steps and materials (such as application of a fluorinated polymer layer as a hydrophobic patch) should be employed in construction of the valve.

This type of valve is compatible with centrifugal microfluidics.

#### 2.1.9. Vacuum Compression Valve [29]

In vacuum compression valving (VCV), the fluid flow into the chamber is restricted due to the absence of the vent hole of the chamber. Consider a design of two fluidic chambers connected by a channel. The upstream inlet chamber (which is closer to the center of the disc) is connected to the downstream outlet chamber (which is further away from the disc’s center). Normally, if both chambers have vent holes, the centrifugal force on the spinning disc will force fluid from the upstream chamber to flow into the downstream chamber. However, this fluid flow can be prevented by two methods [27]: if the vent of the upstream chamber is blocked (so-called vacuum valving), the fluid attempting to go into the downstream reservoir will cause reduced pressure in the upstream chamber, creating suction that will retain the fluid in the upstream chamber; if, on the other hand the vent in the downstream chamber is closed (so-called compression valving), then fluid from the upstream chamber can be transferred into the downstream chamber because of the air trapped and then compressed in the downstream chamber.

If the centrifugal or burst pressure is known, we can calculate the burst RPM for a VCV by using the following formula:(7)$$rpm=\omega \;x\;\frac{30}{\pi }=\sqrt{\frac{P_{centrifugal}}{\rho \Delta r\bar{r}}\left(\frac{30}{\pi }\right)}$$
where ω is the angular velocity in radians/s, $P_{centrifugal}$ is the pressure needed to burst the valve, ρ is the density of the fluid, Δr is the difference in the radial location.

The design of VCV is incorporated into the CD manufacturing process. The design used in the publication [66] utilizes a 4 mm thick acrylic plastic that has chambers cut into it, covered with a pressure-sensitive double-sided adhesive that is covered by another, thinner 2 mm thick acrylic disc containing the vent holes. The vent holes after disc construction are blocked with wax. The author measured the burst frequency for a vacuum and compression microvalve. For the identical design, the vacuum valving required higher RPM to burst the valve than the compression valve.

The main advantage of this microvalve is the simplicity and, similar to other passive valves, lack of the need for external actuating mechanisms, which makes it affordable and simple to use. The major disadvantage is the use of wax, which can cause contamination of the samples if not handled properly. This issue can be avoided by not drilling the vent holes in the upstream or downstream chambers. However, the reliability of such valves will be affected (similar to all passive valves) by manufacturing variations, as well as by changes in the surface properties of the plastic, which are affected by the shelf life and storage conditions.

As the VCV is designed to use the centrifugal-based pumping mechanism, it is compatible with it.

#### 2.1.10. Passive Liquid Valve [30]

The passive liquid valve (PLV) [21] valving technology is an extension of the vacuum compression valve [27] discussed above where, instead of a wax plug, the liquid filling the venting chamber is used to control the venting to the upstream or downstream chambers. Three different parameters that affect the performance of the developed PLV (i.e., liquid height in the venting chamber, liquid density, and venting chamber distance from the CD center) were characterized and compared with the theoretical results. The microfluidic liquid switching and liquid metering processes were performed utilizing the passive liquid valve. This valve also uses the formula in Equation (7), which determines the valve burst rpm.

The passive liquid valve is simple and inexpensive (the fabrication cost is around 10 cents). The disadvantages of PLV are those shared with other passive valves—the disc-to-disc variation of burst frequencies due to plastic surface property changes and due to manufacturing variations. The PLV is designed to be used on CD platforms.

#### 2.1.11. Water Clock Microvalve [31]

The water clock microvalve, a sophisticated variation of the PLV, was first introduced by Ukita et al. in 2015 [31]. There are two major designs variants of this microvalve: a base clocking chamber-based design (Figure 6a) and a serial chamber-based design (Figure 6b). The base clocking chamber design consists of a clocking chamber filled with fluid and vent channels connected at various levels of the clocking chamber. The clocking chamber has a vent at the top; when the fluid level decreases below the level where the vent channel is connected, the channel vents to the atmosphere and the fluid placed in the chamber that now has unobstructed access to the vent starts emptying into the escape chamber.

In the serial design, the vent channel is connected serially through the sample chambers. Once one sample chamber empties, it opens access to the venting for the next sample chamber, which can empty now, and the dispensing from the sample chambers proceeds sequentially. Both the angular velocity of the CD and the geometry of the fluidic network can be designed for a particular dispensing sequence of the samples.

This design is especially appropriate for serial dispensation of various samples and/or reagents for various assays on CD. At the same time, this valve design occupies significant real estate on the disc and cannot be used as a substitute for the simple on–off valves that are used most often.

Given the complexity and sophistication of the possible samples/reagent dispensing programming that the user can implement, the extra fabrication cost is insignificant and is evaluated to be around 10 cents per disc, consistent with other passive valves. 

#### 2.1.12. Aliquot Microvalve [40]

The aliquoting microvalve (also known as the centrifuge-pneumatic valve) was introduced by the Zengerle group [40]. There are two main types of such a microvalve: the two-stage design and one-stage design. In the two-stage design, there is a main distribution channel with a number of separate metering chambers radiating from the feed channel as presented in Figure 7. There are hydrophobic patches at the bottom of metering chambers leading to unvented reaction chambers. In the first stage of the operation the fluid flowing through the feed channel fills the metering chamber, then in the second stage of the disc operation, the angular velocity of the disc is increased significantly to cause bursting of the hydrophobic valves at the bottom of the metering valves and the reaction reservoirs are filled.

In the one-stage design the radially positioned reservoirs are connected to the feed channels by small capillary channels that act as capillary burst valves.

The major advantage of this microvalve is that it offers precise aliquoting of the sample volume into smaller volumes. The disadvantage of the valve is high tolerances that are required for executing this design and the additional cost of hydrophobic patches if the two-stage design is used. These valves are designed to be used in cases where the fluid is transferred into a “dead-end” reservoir that will not have channels running out of it. thus, only having one inlet.

The aliquot-based microvalve is compatible with CD fluidics, and the cost is similar to CBV and hydrophobic valves.

#### 2.1.13. Suction-Enhanced Capillary Microvalve [35]

Suction-enhanced capillary microvalves, first introduced by Gorkin et al. [35], are based on non-linear pressure gradients generated in radially located channels connecting two fluidic chambers, and there is a channel from another chamber that joins that connecting channel at the T-junction (see Figure 8). When the fluid runs in the connecting channel, the dynamic pressure in that channel drops, creating the suction that allows the fluid from the adjoining secondary reservoir to be transferred over the siphon crest and into the downstream waste reservoir. The authors presented a theoretical treatment of the pressure variation in the fluidic network during the disc’s operation.

We derive the equation that determines the pressure variation with the length of the channel as a parabolic function as follows: (8)$$p\left(z\right)=p_{o}+\frac{1}{2}\rho \omega^{2}\left[z^{2}-\left(\frac{R_{1}{}^{2}-R_{0}{}^{2}}{L}+L\right)z+\left(R_{1}{}^{2}-R_{0}{}^{2}\right)\right]$$
where *p*(*z*) is the final pressure at the z location in the channel, (*z* is the length dimension of the channel), $p_{o}$ is the pressure at the base of the channel, *R*_0_ is the radial location of the fluid level in top chamber, *R*_1_ is the radial location of the base of the channel, and *R*_2_ is the final location of the bottom of the channel.

The advantage of this type of microvalve includes the flexibility in the bill of materials that can be used for the disc, since this valve that creates suction allows the use of the hydrophobic materials. The disadvantage of this valve is the need for additional real estate on the disc to include the additional upstream (loading) reservoir.

Suction-enhanced capillary microvalves are appropriate for use in centrifugal microfluidics, and the cost is comparable with the other passive valves discussed above.

#### 2.1.14. Dissolvable Film Microvalve [37]

The dissolvable film microvalve [37] was introduced by Gorkin et al. in their 2012 paper. In this valve, mainly designed for a centrifugal platform, a dissolvable film valving tab consisting of the dissolvable film (DF) (made of an aqueous polymer matrix of a range of cellulose, hydrocolloids, acrylate copolymers, gums, polysaccharides, and plasticizers) is attached to a pressure sensitive adhesive with a hole in it and stacked on top of the valving channel opening. When the liquid is centrifugally pumped into the valving area by spinning the CD, the absence of venting traps the air and prevents fluid from entering the DF-covered hole (similar to the vacuum compression valve discussed above in Section 2.1.9). To open the valve, the spin rate of the CD is increased to allow some of the fluid to enter the chamber and gradually dissolve the film (in 10 s) so that the vent hole opens up and the fluid can proceed to travel downstream into the outward chamber. The authors also found that the increase in burst frequencies correlates with the increase in the DF film dissolution time.

The main advantage of the DF microvalve is its simplicity and flexibility in achieving various burst frequencies based on selecting specific channel and reservoir geometry. The disadvantages include added manufacturing steps and the use of non-standard materials that might also be affected during the storage.

This valve is compatible with CD fluidics, and the cost for the DF includes the fabrication of the film and installing it on CD.

#### 2.1.15. Environment-Responsive Valves [36]

Increasingly, valves used in Lab-on-CD use environment-responsive materials. For example, the concentration valve [33], composed of a synthetic gel that swells in response to the glucose concentration, was first demonstrated for use in detecting changes in glucose concentration. This glucose-concentration-sensitive gel is synthesized from poly(N-isopropylacrylamide) (PNIPAAm) polymer derivatized with a fraction of phenylboronic acid, which senses the glucose when a small amount of cross-linking agent in the form of N,N’-methylene-bis-acrylamide (NMBA/NB gel) is used. The authors demonstrate that swelling temperature increases from 22° to 36 °C as the concentration of glucose varies. Thus, the gel responds to fluctuations in glucose concentration. There are numerous examples of other hydrogels that swell/shrink in response to other environmental factors such as temperature or acidity [67,68,69,70].

The major advantage of this microvalve is the responsive nature of the valve actuation that allows for a great degree of flexibility and sophistication in design of the assays or microfluidic reactors on the disc. However, it is difficult to guarantee the reliable operation of such valves, and their actions are affected by time of storage and change in environment during the storage and operation of the disc.

In principle, the environment-responsive valves are compatible with the CD design. For example, the gel that swells or shrinks can be contained in a cage within the microfluidic channel. The fluidic path is open when the gel is in a shrunken state and the fluidic path is blocked when the gel is swollen. The manufacturing cost for such a valve can be fairly high due to the complex fabrication route (microfabricated cage, for example), as well as the cost of environment-sensitive gel.

### 2.2. Active Microvalves

Active microvalves require the application of external force (other than the centrifugal, Euler, or Coriolis forces intrinsic for CD platforms) for actuation of the valves. These external actuators for turning on or off of the active valves are various and include laser [45,46,47,66], magnetic actuation [15,48,71], mechanical actuation [49,50,51,52], phase-change-based actuation [55,56,57,58,60,62,63], electrical actuation [16], thermal actuation [17,61], chemical actuation [72], and pressure actuation [12,64]. Generally, active valves are more reliable than passive valves due to the active valves’ smaller degree of reliance on the precision of machining, fabrication, and constancy of material properties compared to passive valves. However, this comes at the expense of a typically higher cost, more complex design, and often leads to larger footprint and larger weight of the test platform compared to platforms that use devices that utilize passive valves only.

#### 2.2.1. LaserPacks [44]

LaserPacks were first introduced in the BioMEMS laboratory at the University of California, Irvine. Their design consists of a sealed mylar-based capsule filled with liquid. The packages are made in-house by thermoforming the Mylar into containers that can hold fluids. These capsules are placed into chambers on a CD. The laser shining onto the black package transfers enough energy to the package to melt the mylar and allow the fluid to exit the capsule.

The major advantage of this microvalve is that the hermetically sealed capsule can be prepackaged for dispensation of reagents on the CD platform. Two main disadvantages of this approach are that it cannot be used for applications that are normally on and then need to be off (or that require multiple stops and release of reagents) and that this technique cannot be used with heat sensitive reagents.

These LaserPacks are compatible with a CDs. The cost of such packs depends on the cost of mylar packaging and access to thermoforming equipment but is likely less than 1 USD per Mylar capsule. Their fabrication and operation of the LaserPacks is presented in Figure 9.

#### 2.2.2. Laser Printed Valve [45,46]

Laser printed valves are printed with black ink on a transparent sheet of a polyethylene substrate. The black ink absorbs more heat than the surrounding transparent plastic and melts preferentially, opening the flow of the fluid through the microchannels on the CD. The response time of the laser printed valve varies depending on the material and thickness of the transparent plastic and the power of the laser.

The major advantage of this microvalve is that the fabrication process is straightforward, but the disadvantage of this microvalve includes additional assembly steps, the inability to use multiple on/off transitions or normally on operation where the fluid flow should be stopped on demand.

#### 2.2.3. Foam-Based Valve [47]

In a paper introducing foam-based microvalves [47], the authors described three mechanisms to close the microfluidic channel on a microfluidic platform—an expanding foam that overflows its reaction chamber and blocks the valving channel and two types based on a modified print-cut-laminate (PCL) adhesion method. In detail, the first (a) method is based on mixing a DABCO (1,4-diazabicyclo [2.2.2] octane) catalyst and a diisocyanate/diol mixture that produces polyurethane foam (FX Supply, A-B foam 2-lb density). This foam overflows the reaction chamber exiting into the main flow channel where, upon hardening, it closes the fluidic path. In the other two methods (b and c), a modified PCL layer is used that has 5 layers—layer 1 is PET (polyethylene terephthalate), layer 2 is PET covered with heat-sensitive adhesive (HAS) on both sides, layer 3 is PET covered with xerographic toner on both sides, layer 4 is the same as layer 2, and layer 5 replicates layer 1. In method b, a spot-welding technique is used to apply heat and pressure to the channel connecting the inlet and outlet with a pin-like structure and irreversibly joins the plastic monolayers. In method c, chloroform is applied to the xerographic layer, and it dissolves the toner and joins them together to seal the channel. The schematics of all the three types of valves are shown in Figure 10.

The major advantage of the foam-based microvalve is that it creates a robust high RPM resistance closure of the valving channel. For layer-based valves, the advantages include the robust open to close and close to open transition. The major disadvantage of these microvalves is the use of external agents to activate and form the sealing element.

The application of centrifugal force is not mandatory for the operation of this microvalve, but there are no elements of the valves that would be incompatible with the CD operation. The cost of the valve depends on whether type a, b, or c is used. Typically, type a will have an easier fabrication structure as it does not require multiple additional layers, but it does require additional reagents. In all cases, environmental variations (temperature, humidity) are expected to affect the reliability of these valves.

#### 2.2.4. Wax-Based Valve [62]

In the wax-based valve, the phase change in wax into a liquid form and manipulation by either applying pressure or fluid flow is used. For a normally closed-to-open switch, the wax is applied initially in solid form to block a channel. This is heated by an external heating element to melt the wax. As the fluid flows, the wax moves upstream to a wider channel and is deposited there to open the valve. In the normally open-to-closed configuration, a T-junction is used in which the valving wax is in the third channel connecting the open valving channel. By applying pressure and heat, the molten wax moves to the main channel, blocks the valving channel, and is cooled to solidify the wax and block the channel. By melting the wax again and applying the fluid flow, the wax moves downstream, as in the previous design, into a wider channel and opens the valve. The valve is illustrated in Figure 11.

The major advantage of this valve is that the maximum resistance RPM for the seal is very high, approximately 5000–6000 RPM. The major disadvantage is that it involves the use of wax, which is very hard to handle in molten form.

This valve is compatible with a centrifugal CD and centrifugal force and helps move the valving wax to outward chambers. The cost of the wax is very low, and the overall cost includes additional machining to add channel modifications to include the valve’s functionalities.

#### 2.2.5. Reversible Magnetic Wax Valve [48]

This valve, fabricated on CD, was introduced by Peshin et al. [48] The microvalve consists of a wax chamber connected to the valving channel, which must be sealed to prevent the flow of fluid through it. The chamber is filled with ferrowax (paraffin wax + ferrofluid) and melted. A magnet is used to pull the ferrowax into the channel that is sealed once the wax cools and solidifies. To open the closed channel the wax is heated, liquified, and pushed downstream by the fluid flowing towards downstream direction by the centrifugal force on the rotating disc.

The reversible magnetic wax valve is compatible with centrifugal microfluidics. The advantage of this microvalve is easy fabrication of the valve (the extra component is the solid pellet of wax placed in the wax reservoir) and the availability of multi-use (on–off–on–off-…) operation that could enable the development of more complex assays on a single CD. The disadvantage of this microvalve is the use of ferrowax, which is listed as a carcinogenic material.

The cost of this type of valve includes the cost of ferrowax and ferrofluid and external elements such as a magnet to actuate the valve. The operation of this valve is shown in Figure 12. The response time of this ferrowax valve is 6 s given the angular disc velocity of 6500 RPM.

#### 2.2.6. Soft Diaphragm Valve [15]

In a soft diaphragm microvalve, the author introduced a valve adapter designed to switch off with a push-and-twist type pin that pushes on an elastomeric epoxy-based film. The pin and adapter were made using a 3-D printer. In an automated version, the top of the pin had a groove that could be turned using an automated screwdriver. The simplicity of this soft diaphragm is highly useful in cases with extreme point-of-care situations. The main drawback is the manual operation required for operating the valve, thus, reducing the options for remote actuation of the valve on the spinning disc. A high degree of fidelity in valve construction is also required to prevent leakages. As this is a derivative of the passive liquid valve, we use Equation (7) to define the burst frequency.

The valve is compatible with centrifugal microfluidics. The added cost of the valve includes the addition of elastomeric film and the cost of 3D printed parts.

#### 2.2.7. Layer-Based Diaphragm Valve [49]

In a layer-based diaphragm valve, the researchers used soft lithography to fabricate a network of elastomeric channels that are positioned atop of each other. When the control channel expands under the influence of the forces of the fluid flowing through it, the expanded control channel presses into the neighboring soft fluidic channel and cuts off the flow of the fluid in that channel due to a restricted cross-section area. If the fluid flow in the control channel is reduced, it lowers the compressive force on the underlying fluidic channel and the flow in the fluidic channel is restored. Therefore, the flow in the fluidic channels depends on the flow rate of the fluid in the control channels. The authors also found that curved shapes (instead of edged shapes) are better in helping reliable valving in the fluidic channel. The fluidic and control channels were fabricated from elastomer with a Young’s modulus of approximately 750 kPa.

The major advantage of this valve is that one control channel can actuate multiple valves on a single CD. The major disadvantage is that the CD fabrication is rather complex and involves a multi-layer soft lithography using PDMS or other soft polymers.

The cost of this type of valve involves the potentially significant cost of fabricating the multi-layer elastomer-based fluidic network on a centrifugal platform (in addition to disc fabrication based on the traditional fabrication of hard matter).

#### 2.2.8. Elastomeric Diaphragm Valve [50]

In the elastomeric diaphragm valve, there is a PDMS-based membrane under the fluidic channel. This membrane is actuated by pressure or vacuum so that the membrane moves up and down to either block or unblock the fluidic channel. The authors also studied the fluid pressure needed for flow through this valve under a particular manifold pressure that increases with an increasing opposing manifold pressure. The valve requires placement of a PDMS (Polydimethylsiloxane) membrane and a specialized design of the fluidic channel to allow for actuation by the application of an external pressure.

While, in principle, this can be a simple design, two sources of potential complication are (a) the difficulty in heterogeneous fabrication that involves both hard plastic and a PDMS membrane (increasing fabrication cost and reducing the reliability of leak-free operation) and (b) bringing in the external pressure or other means of forcing the deflection of the PDMS membrane on a spinning disc.

#### 2.2.9. Magnetic Ball Valve [51]

In a magnetic ball valve, the main active component is a ball-shaped magnet that moves inside the fluidic chamber. When the ball blocks the escape channel and the CD spins, the fluid is unable to escape into the escape chamber. Therefore, an external magnetic field is applied to move it out of the valving channel and open it, which opposes the centrifugal force to keep the valve open. This is achieved by putting an array of magnets underneath the rotating CD.

This microvalve is simple and easy to use due to the use of pre-installed magnets. While the valve performed well in laboratory conditions, it is yet to be seen if such a valve with moving parts can perform reliably when tight feature tolerances are more difficult to maintain under conditions of mass-production.

The cost of this valve is equal to the manufacturing of a CD plus the cost of the magnetic ball and an array of actuating magnets. This valve is compatible with a centrifugal platform.

#### 2.2.10. Pneumatic Soft Valve [52]

The pneumatic soft valve is based on the actuation of a simple flexible diaphragm over the valving channel fabricated on a chip or CD. This diaphragm covers the channels for valving and is made with a soft elastomer. After applying pneumatic external pressure, the valve closes and opens when no pressure is applied. This valve is shown in Figure 13.

The major advantage of this valving system is that pressure can control the flow of the fluid through the channel. The major disadvantage of this microvalve is that the valve requires a sophisticated PDMS soft membrane to actuate robustly.

The cost adds up by requiring the cost for fabricating the PDMS/elastomeric soft membrane and a pneumatic external pressure application. This is not compatible with centrifugal fluidics as is and we need to redesign the valve to have a portable pneumatic pressure installed on the CD or a way to automate or apply pressure on the soft elastomeric layer.

#### 2.2.11. Soft-Lithography-Based Pinch Valves [53,54]

In soft-lithography-based pinch valves, the CD or the microfluidic platform and valving channels are made of PDMS elastomer. In fabrication using soft lithography, the pressure applied on the channels varies the flow rates in the channels below the pinching. This means that as higher pressure is applied, the valve switches off, and it switches back on at zero pressure. The external pressure is applied with the help of mechanical actuators attached to the CD. The robustness of the valve due to the stable elasticity of the adhesive layer and the reusability of this valve are its main advantages. The cost of manufacture of this valve includes the soft-lithography-based fabrication of the microfluidic CD and manual operation and pinching of the valves to switch it on or off. This valve is compatible with the CD platform and requires external pinching to switch on/off.

#### 2.2.12. pH-Controlled Hydrogel Valve [55,72]

Per the example of the application of the environment-sensitive smart materials, pH-controlled cylindrical hydrogel valves respond to changes in the pH of the surrounding fluid by swelling or shrinking and, thus, respectively, closing or opening the fluidic channel. To fabricate the cylindrical hydrogel valves, the fluidic channels are filled with a photopolymerizable liquid consisting of acrylic acid and 2-hydroxyethyl methacrylate (in a 1:4 molar ratio), ethylene glycol di methacrylate (1 wt%), and a photo initiator (3 wt%). This liquid is exposed to UV light under a mask, and when the polymerization is finished, the channel is flushed with water to remove the unpolymerized liquid. The authors also demonstrated that smaller diameter cylinders of hydrogels respond to pH changes faster than bigger diameter cylinders due to a higher surface to volume ratio of the smaller hydrogel cylinder.

The major advantage of this valve is its sensitivity to environmental factors (pH), but drawbacks include insensitivity to other parameters and inability of the valve to be actuated by other means (on an as-needed basis). Additionally, the complexity of the fabrication and assembly of this valve will increase the cost of production (we are unable to estimate the specific increase in cost of the platform when such a valve is utilized).

These hydrogel valves are compatible on a CD platform and the main cost incurred is the cost of fabrication of these hydrogel cylinders.

Another example is the case of the hydrogel shrinkage pH-based valve; when the hydrogel sensitive to the pH value of the fluid passing is used; it either stops the backflow or allows forward or backward flow. The pH-sensitive hydrogel strips are made via a combination of parallel and sequential photopolymerization. When exposed to a high pH fluid, the pH-sensitive strips swell while pH-insensitive strips remain unchanged; this causes the bi-strip to bend and close the channel, acting as a check valve. In accordance with the orientation and design of the bi-strip, the forward flow pushes the bi-strip apart and allows the flow while backward flow is not possible. The total time to fabricate this valve is 10 min. It is compatible with a centrifugal platform, but performing constant pH assays is impractical, and it requires pH variation to operate.

#### 2.2.13. Thermally Soluble Polymer Valve [56]

The thermally soluble polymer valve is another example of the environment-sensitive valve. It is made of a polymer that expands or contracts based on the temperature of the fluid flowing past it. Thus, when the fluid cools down, the valve closes, and when the fluid is heated, the valve opens again. The valve is made of monolithic plugs of poly(N-isopropylacrylamide) crosslinked with 5% methylenebisacrylamide by photoinitiated polymerization. To cool and heat the fluid, the microfluidic platform needs external thermoelectric elements. The response times for this valve are: to open, the valve requires 3.5 s; and to close, the valve requires 5.0 s. This valve is compatible with a centrifugal platform, and the cost consists mainly of the cost to fabricate it using photoinitiated polymerization.

#### 2.2.14. Gel-Based Valve [57]

In the gel-based valve, the light-sensitive gel contracts in the presence and expands in the absence of light. The gels are made by free-radical copolymerization of N-isopropylacrylamide (main constituent) and light-sensitive chromophore trisodium salt of copper chlorophyllin. This valve is compatible with the centrifugal platform and can be easily actuated by using a partially transparent adhesive to control the amount of light exposed onto the valve. The main cost of this valve is fabrication of the light-sensitive gel.

#### 2.2.15. Temperature Memory Gel Valve [58]

The temperature memory gel valve is made up of two types of gel—NIPA or N-isopropylacrylamide, which is sensitive to temperature, expanding at high temperatures, and PAAM gel or acrylamide gel, which is sensitive to the concentration of the acetone solvent. They are combined in a bi-gel strip by making the NIPA gel and PAAM gel separately and then allowing the PAAM gel to diffuse into the NIPA network for one hour before initiating polymerization to make NIPA-PAAM interpenetrating networks (IPN). The final product is 2–3 mm thick IPN, which is sensitive to temperature. The valve opens at 38 °C and closes at 22 °C.

This valve is compatible with the centrifugal platform and needs actuation by either contact or remote heating applied to a spinning CD to open and close the valve. The main cost of this valve is the fabrication of a layered gel.

#### 2.2.16. Thermally Actuated Membrane Valve [59]

The thermally actuated membrane valve uses the property of the olive oil to expand when heated and a pneumatic expansion to block the valving channel, as shown in Figure 14. The CD consists of a simple inlet chamber, outlet chamber, and a connected channel. The layers are three in number and made up of PDMS (polydimthylsiloxane) soft elastomer. The first layer consists of the chambers and channel. The second layer consists of the activating layer, on which the oil pushes to close the valve. The final and third layer consists of the ports and openings for the device to place. The device consists of a simple oil filled elastomeric tube that has a resistance wire and is sealed on the top by epoxy glue. The valve works such that once the resistance element is switched on, it heats up and heats the oil filled around it. Due to high temperatures, the oil expands, and the elastomeric layer of PDMS is pushed and expands into the valving channel, as shown in Figure 14d. This switches the valve off. Switching off the DC power cools down the wire and oil and the PDMS reaches its original shape and opens the valving channel. This switches the valve on.

The valve has a response time of approximately 2 s for valve closing (heating) and 9 s for valve opening (cooling). The authors claim the cost of this whole setup is approximately 1 dollar. As this valve is designed for the centrifugal platform and thus, it is compatible with it.

#### 2.2.17. Solid–Liquid Phase Change Valve [60,61]

The solid–liquid phase change valve is a specially fabricated ice valve in which the solidification of water into ice stops the flow and heating it back to liquid opens the valve. The valve is based on a layered composition of the platform in which layers consist of a thermoelectric cooling device (TECD), a working channel, a fin-like heat dissipater, and an optional electric heater. A positive electric voltage application on the TECD helps freeze the working fluid to ice in a short time. When voltage is applied in the opposite direction, the platform heats the fluid and opens the valve.

The limitation of this microvalve is that it mixes water with the flowing analyte, leading to excess dilutions. This valve is compatible with a centrifugal platform. Fabricating the CD with the Peltier element is the main cost of application of this valve.

#### 2.2.18. Vacuum Wax Valve [63]

In the vacuum wax valve, the authors designed a T-junction with the attached channel to the fluidic inlet and outlet ports. The attached channel connects to a pressure port, which is also laid with heaters, and wax enters the fluidic channel through this port. As the wax is loaded and melted, the pressure is applied to push the molten wax into the T-junction, closing the fluidic channel. To open the valve, vacuum (negative pressure) is applied, which withdraws the wax in the T-junction back into the attached channel, which opens the valve.

The velocity of the molten piston in the stem channel is related to the applied pressure using the equation:(9)$$v=\left(\frac{d^{2}}{S\mu L}\right)\Delta P$$
where *v* is the average drop velocity, *d* is the channel depth, *S* is the geometric shape factor, *µ* is the bulk viscosity, *L* is the piston length, and ΔP is the internal liquid phase pressure difference between the drop ends.

This specialized valve is applicable on a centrifugal platform and needs specialized pressure and vacuum ports to operate the valve (major disadvantage). The cost of this valve includes the cost of fabricating portable vacuum and pressure ports and the cost of wax. This valve is illustrated in Figure 15.

#### 2.2.19. Electrically Controlled Valve [16]

In the electrically controlled valve, which was first presented in 2008 [16], the researchers use the property of polyethylene glycol (PEG) to change the volume substantially upon phase transition from the solid to the liquid phase (and vice versa). The valve is activated via heating by resistive elements near the PEG to change its phase. The structure of such a valve consists of a glass outer structure, PEG inside the glass, and covered by a PDMS flexible membrane. When the temperature is below 40 °C, the PDMS layer contracts due to less space occupied by solid PEG, which opens the valve. However, as the temperature elevates above 50 °C, the PEG liquifies and occupies more space, pushing the PDMS membrane to close the valve. This valve is compatible with a centrifugal platform and supports the motion of fluids using centrifugal propulsion. The cost of the valve consists of the cost of purchasing the PEG and fabricating its seating chamber to operate the valve, in addition to microfabricating the resistive heater or purchasing as is from an external manufacturer.

#### 2.2.20. Air Pressure-Based Check Valves [64]

In air pressure-based check valves, the inlet and outlet are connected by a channel covered with a latex membrane. As shown in Figure 16, there are two design variants—(a) the latex membrane covers the inlet port and has a hole in it. When the fluid flows in the forward direction, the latex expands and allows the fluid to flow. However, reverse flow creates negative pressure, and the membrane blocks the flow. In variant Figure 16b, when there is fluidic pressure in the forward direction, the latex is stretched, and the valve works fine and stays open. If we apply negative pressure in the inlet port, the latex covers the channels and does not allow the fluid to flow. Thus, the channel is closed off.

Once the actuation pressure is determined experimentally, the deflection of the latex film can be calculated using the following equation:(10)$$u=\frac{3pa^{4}}{16t^{3}E}\left(1-v^{2}\right)$$
where *u* is the deflection of the latex layer, *p* is the pressure applied across the latex layer, a is the radius of the chip, *v* is the Poisson’s ratio, *E* is the Young’s modulus, and t is the thickness of the latex layer.

This valve is compatible with a centrifugal platform. The main cost to fabricate this valve is the cost to assemble the latex inside the CD.

#### 2.2.21. Xurography-Based Valve [12]

The Xurography-based valve uses a basic principle of the VCV, as described in Section 2.1.9, and a dissolvable film valve, as in Section 2.1.14. It is an active valve and is actuated when the vent hole is opened using the external force. In detail, a venting principle is used to open the valve by piercing the vent’s pressure sensitive adhesive (PSA) covering, which opens the flow to a dissolvable film (DF), wetting it to allow the fluid to flow out and open the valve. The chamber has a u-shaped channel where the fluid enters due to centrifugal force, and as it is not vented to the atmosphere, it compresses the air. The parameters are designed such that the fluid stops just before the channel to the DF valve. To open the valve, the PSA vent covering is pierced by a cutter plotter, and this allows the compressed air and fluid to move out and enter the DF channel. The wetted DF allows the fluid to flow. This valve is designed for a centrifugal platform by the author and is, therefore, compatible. The main cost of this valve is installation of the cutter plotter machine and fabrication of the DF channel.

#### 2.2.22. Hydrophobic Laser Printer Lithography Valves [65]

In the hydrophobic-patch-based laser printer, lithography microvalves use a multi-layered fabrication of the CD. It consists of three layers of the Pe (Polyethylene) film. The black printer toner is directly patterned onto the surface of polyester film, enabling Poly (ethylene terephthalate)-Toner (PeT) on the top and bottom of the CD, which is cut to create a fluidic channel. Once the printed toner-based patch is applied onto the channel (ablated using the CO_2_ laser), it creates a hydrophobic patch that repels fluid. This toner patch was shown to have a lower contact angle of 51° from 111°, thus helping the hydrophobic patch property at the location of the toner print.

This valve is designed for a centrifugal platform and is, therefore, compatible with CD operation. The main cost of this valve is installation of the laser cutter and printer machine and fabrication of the laser ablated channel.

### 2.3. Hybrid Microvalves

Few valves can be considered hybrid microvalves—valves that are neither active nor passive but that, to some extent, belong to both categories. Prominent among them are valves such as the capillary-driven wax valve [13,48], actuated by an active thermal phase change and operated by passive capillary forces, and LIFM [14,17,66], a single-use valve that is also a wax-based phase change—actively by thermal action and passively by expansion of molten wax. We will discuss both types of valves below.

#### 2.3.1. Capillary-Driven Ferrowax Valve [13,48]

This valve, introduced in 2021 [13,48] (see Figure 17) uses the capillary motion of the ferrowax, after which the wax melts via external heating (laser or hotplate). The main feature of this valve is that it can be switched from on to off mode and from off to on multiple times on a centrifugal platform. The valve consists of two chambers, one radially inward chamber (closer to the disc’s center) and one radially outward chamber (further away from the disc’s center) connected by a fluidic channel. This channel is connected to a trapezium-shaped wax inlet chamber. All the chambers are sealed with a single-sided pressure-sensitive adhesive. When the wax is filled in the inlet wax reservoir and actuated by a laser to melt by capillary motion, it moves into the connecting channel to finally solidify by cooling, blocking the channel and switching it off. This is called a manufactured closed valve. When the valve must be switched off, the blocking wax is simply melted, and the disc is spun to throw the molten wax into the radially outward chamber, clearing the valving channel for fluid flow, thus, opening the channel.

Here, the capillary flow length is proportional to the root of time and can be calculated using the following equation:(11)$$L=\sqrt{\frac{\gamma ht\;cos\theta_{c}}{3\mu }}=W\sqrt{t}$$
where *W* denotes the Washburn constant and is a function of surface tension γ, the viscosity of the fluid μ, contact angle of the fluid with channel walls $\theta_{c}$, and the depth of the channel h.

As the opening of this valve uses centrifugal force, it requires use on a centrifugal platform. The cost of this valve involves machining/molding the wax inlet chamber and applying the ferrowax.

#### 2.3.2. LIFM (Laser-Irradiated Ferrowax Microvalve) [14,17,66]

The laser-irradiated ferrowax microvalve (LIFM) was first introduced in 2007 [15]. The valve has two design and operation processes—one for the closed to open mode switch and the other for the open to closed mode. Both are actuated by a laser source and use ferrowax as the means for blocking the valving channel. The common base design has a channel connection through which the fluid flows. In the first design (also called normally closed (NC)-LIFM), there are two chambers along the valving channel with a thick channel connecting them where the ferrowax is applied to block the channel. The mode switch occurs when the ferrowax location is heated by the laser, the wax melts and, due to the fluid flow, moves to the subsequent chamber, where it collects. As soon as the ferrowax moves, the channel is now in the open state. In the second design (also called normally open (NO)-LIFM), instead of two consecutive chambers, the channel is connected to a single wax chamber that is filled with wax. When the wax is melted by laser, due to expansion and phase change, the molten wax moves into the channel and closes it after cooling it. Thus, this closes the valve from its normally open state. The author also explored the effects of ferrofluid volume fraction in the ferrowax and laser power on the valve response time. Summarizing, the results show that as the volume fraction of the ferrofluid increases, the response time decreases due to the lowering of the melting point of the ferrowax. Higher laser power leads to a lower response times. This valve is compatible with a centrifugal platform. Similar to the Capillary-based ferrowax valve described in Section 2.3.1, the cost of this valve involves the cost of the ferrowax and the cost to fabricate the chambers for the valve functioning.

## 3. Discussion

### 3.1. Fabrication Techniques

The cost and reliability of various valves described in the present survey depend in a significant way on the fabrication of the lab-on-a-disc in general and each kind of valve specifically. Table 1 summarizes the fabrication techniques for manufacturing the discs and various types of valves. Most of the passive valve fabrication is based on laminated object manufacturing (LOM) [73], based on subsequent stacking of the layers that are glued/bonded to each other. Typically, when these types of lab-on-disc go to commercial production, injection molding in combination with ultrasonic bonding replaces CNC milled acrylic sheets (or other polymer layers) that are joined together with interspersed double-sided adhesive layers [74]. Active valves add various other manufacturing approaches that include methods such as hydrogel assembly, soft lithography, assembly of magnets, and other techniques, and additional parts and materials such as heating/cooling elements (for example, Peltier elements) are incorporated in the disc fabrication. It is not unusual in the fabrication sequence of active valves to have combinations of the two or more fabrication steps, such as multiple steps in patterning via photo-polymerization of the hydrogel valves (see Section 2.2.14 and Section 2.2.15 above).

### 3.2. Robust Operation of Valves

In Figure 18, we plotted the various types of active and passive valves on a graph with maximum resistance RPM (in revolutions per minute) on the y axis and response time (in seconds) on the *x* axis. Here, we define maximum resistance RPM on a CD as the maximum pressure that the closed configuration of the valve can withstand before bursting and allowing the microfluidic flow. In order to normalize the expected operating parameters of various valves discussed in this study, we calculated the critical spin rate of CDs (in revolutions per minute), assuming the same location for all valves to be 20 mm away from the center of a CD with a disc diameter of 80 mm.

We noticed that passive valves respond quickly to the change in spin rate (most of the passive valves have a response time of a few seconds) with a bimodal distribution of critical rpm in 1000–2000 RPM and 4000–6000 RPM regions. Thus, if we need a low-pressure application and a low response time, we can choose from the passive valves in the region of 1000–2000 RPM, which are hydrophobic [38,39,40], pressure-based [29,31], and capillary-based [32,33,34,35] valves. Additionally, for a low response time and high fluidic pressure application, we can choose from valves in the range of 4000 to 6000 RPM—the siphon-based [41,42,43], spin-based [11,18,19,21], and dissolvable films [37].

Active valves, on the other hand, present a wider range of options than passive valves—both in enabling operations at higher rpm and also usually requiring a longer response time than passive valves. For applications requiring low resistance pressure and low response time, pressure-based [64] and chemical-based [72] valves can be used. For high resistance pressure and low response time applications, we can use laser [45,46,66], magnetic-based [48], and mechanically actuated [49,50,51,52,53,54] valves. While the applications for active and passive valves fall in almost similar ranges, in the case of phase change active valves and electrically actuated valves, response times are longer.

The most important consideration for utilization of the active valves is the degree of robustness that is expected. For critical applications (including health tests), it is more usual to select more reliable active valves, even if the cost is higher and response time is longer than for the passive valves.

## 4. Conclusions

The centrifugal microfluidic platforms present a number of advantages for a wide range of complex chemical and biological assays, but reliable operations of CD fluidic devices require precise and robust valving capability. This critical review presents a comprehensive list of the available microvalves reported in scientific publications.

We have classified the valves into active, passive, and hybrid categories based on their actuation mechanisms, and we detail the actuation mechanism for each valve type. We analyze the compatibility of the valves to application on centrifugal platforms and report (whenever known or possible to evaluate) the added cost to implement these valves.

Passive valves on CDs are actuated by the change in spin rate, spin direction, or spin acceleration of the disc and the critical burst frequencies of these valves are controlled by the geometry of the fluidic network on the disc and by physio-chemical properties of the materials of the disc. Consequently, changes in the material properties of the disc (for example, due to exposure to air, changes in humidity, etc.) or minor changes in channel geometry (due to manufacturing imperfections) affect reliability of the action of passive valves. On the other hand, the action of the active valves relies on the external application of force and energy to activate the valve. Many forms of external actuation include magnetic force, laser, other types of heaters, pneumatic force, etc. The hybrid valves feature elements of both active and passive valves; for example, an external heat source melts the wax, which is then carried by a capillary force towards the fluidic channel that is blocked when the wax solidifies.

We hope that by reviewing the array of available options for the valves on centrifugal platforms, the reader will be able to select the type of valve most appropriate for a specific spin regime and for the required application of an assay under development. The cost and portability considerations favoring passive valves in many cases outweigh the reliability concerns favoring active valves. It is possible that when active valves will be developed to be even less expensive to implement and when external actuation mechanisms (for example, lasers) will not add much to the cost or weight of the platform, we will see wider proliferation of the active valves on centrifugal assay platforms.

## 5. Patents

A patent was filed at UC Irvine under patent number USSN: 63/176,063 for the device in Section 2.2.1.

## Figures and Tables

**Figure 1 sensors-22-08955-f001:**
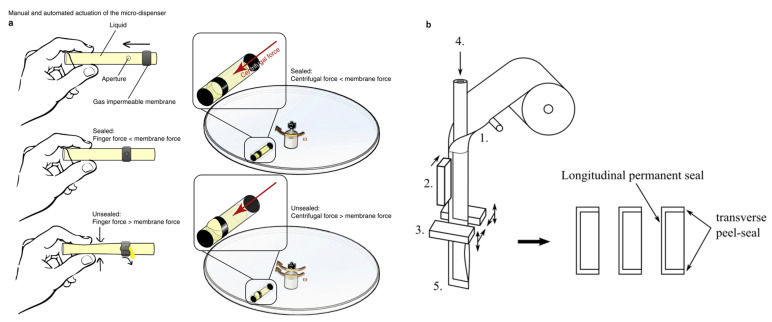
(**a**) Design of the “Flexpenser” pouch-based valving system, showing its internal structure and dispensing mechanism. (**b**) Design of stickpacks. 1. Forming of a tube of the composite foil. 2. Longitudinal permanent sealing. 3. Transverse peel-sealing. 4. Pump-induced dispensing of liquids. 5. Cutting. Source: Reconstructed from publications [11,23]. Permission granted by the publisher.

**Figure 2 sensors-22-08955-f002:**
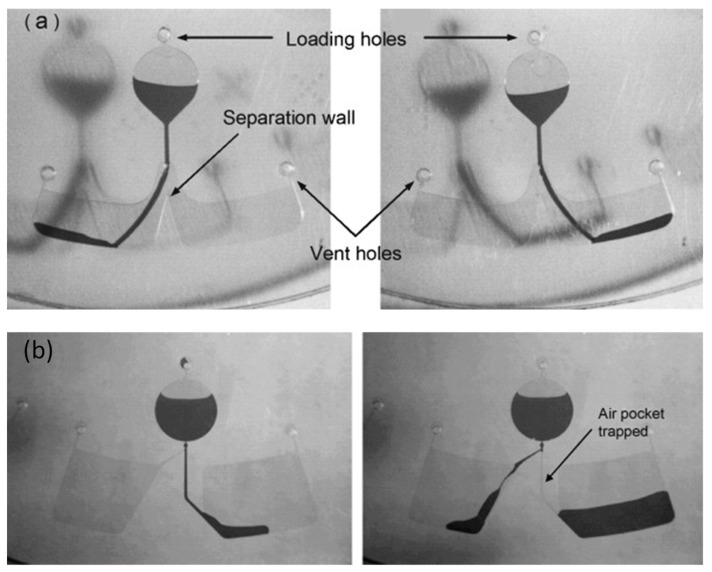
(**a**) Coriolis switch valve [27], in which Coriolis force is the driving force; spin direction switches the fluid flow between the two chambers. (**b**) The air plug valve [27] shows that the air trapped in the wider channel causes the switch of the fluid flow from the wider channel on the right side to the tilted narrower channel on the left side. Source: Reconstructed from publication cited [27]. Permission granted by the publisher.

**Figure 3 sensors-22-08955-f003:**
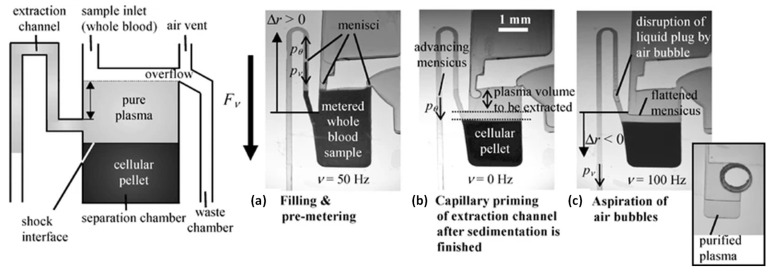
Sketch of the plasma extraction structure. (**a**) After injection through the inlet, a droplet of raw blood is pre-metered. (**b**) The shock interface separating pure plasma and cellular blood proceeds and stops at a position radially outwards compared to the extraction channel. Then, the disk is stopped, and the extraction channel is filled by the capillary pressure p_θ_. (**c**) If the net radial length Δr between the downstream meniscus and the liquid level in the separation chamber is negative, a centrifugal pressure p ν exists to pump the plasma through the extraction channel until air is sucked into the extraction channel. The extracted plasma is collected in a reservoir attached to the extraction channel for further use. Permission granted by the publisher [43].

**Figure 4 sensors-22-08955-f004:**
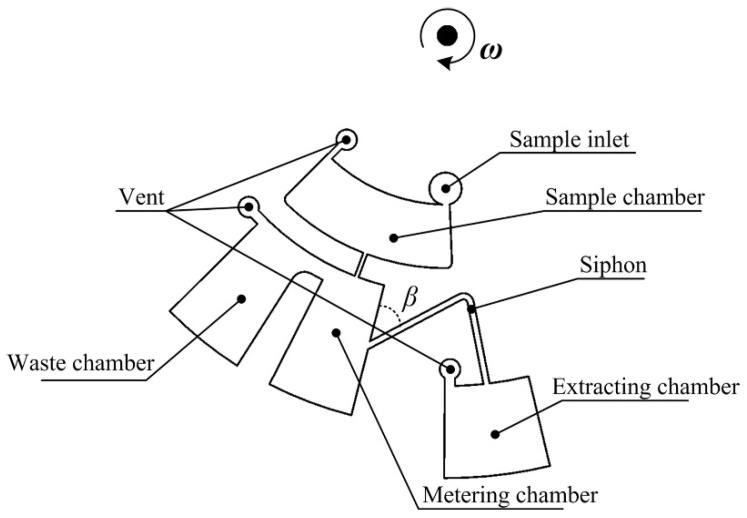
Schematic of the Euler siphon valve on a CD microfluidic chip, where β is the inclination angle of the siphon channel. Source: Reconstructed from publication cited [28]. Permission granted by the publisher.

**Figure 5 sensors-22-08955-f005:**
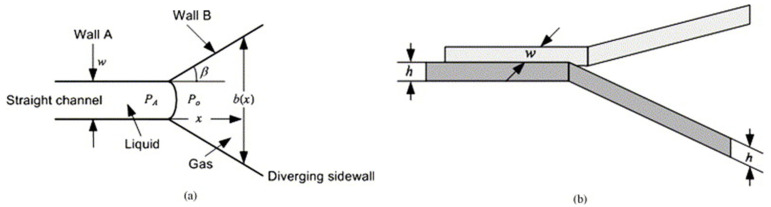
The capillary burst valve in a rectangular channel. (**a**) Top view. (**b**) Bird’s eye view where the h is the height of the channel and w is the width of the channel Permission granted by the publisher [32].

**Figure 6 sensors-22-08955-f006:**
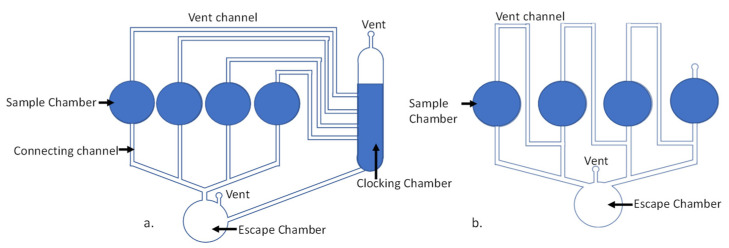
(**a**) The clocking chamber-based design. (**b**) Serial chamber-based valve design. Source: Reconstructed from publication cited [31]. Permission was granted by the publisher to reproduce.

**Figure 7 sensors-22-08955-f007:**
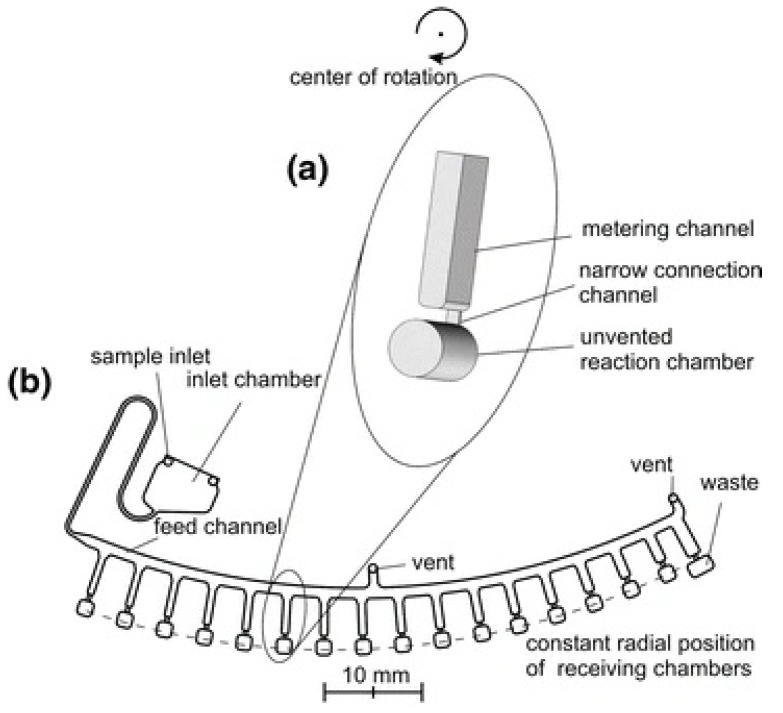
Layout and functional principle of the centrifugo-pneumatic valve and the aliquoting structure. (**a**) The 3D image shows a view on a single metering structure. (**b**) A radially inclined feed channel supports several branching metering structures. Permission granted by the publisher [40].

**Figure 8 sensors-22-08955-f008:**
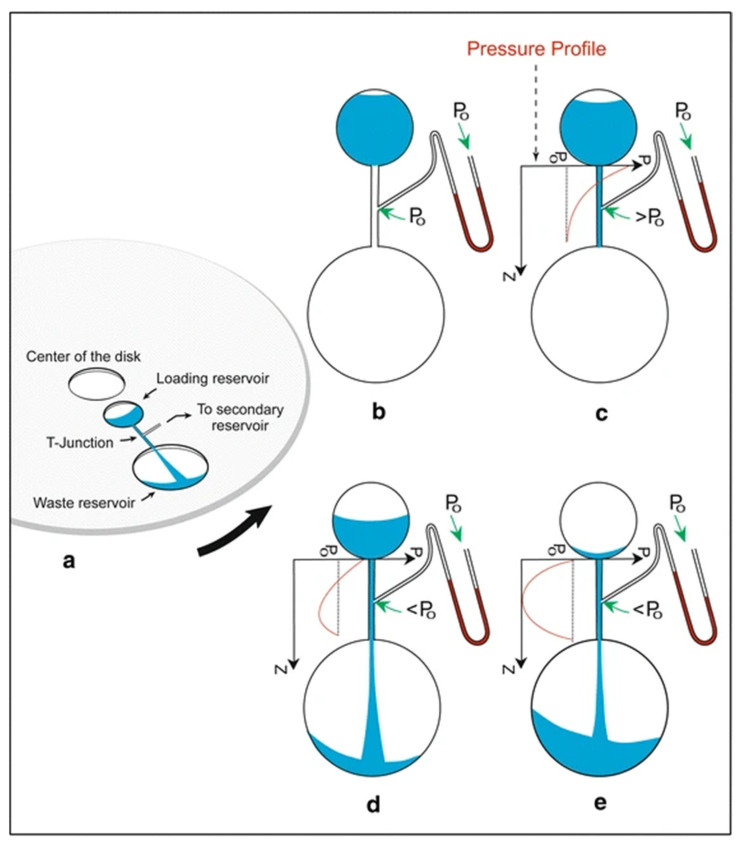
Schematic demonstrating the action of the suction-enhanced capillary valve on a centrifugal platform. (**a**) A view of the centrifugal microfluidic structure; (**b**–**e**) pressure profile changes in the microchannel during emptying (P_0_ is an atmospheric pressure). Permission granted by the publisher [35].

**Figure 9 sensors-22-08955-f009:**
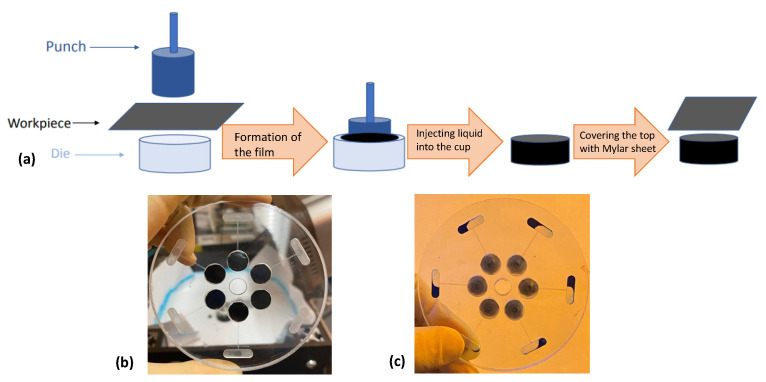
(**a**) LaserPack [44] fabrication using thermoforming of the mylar film. (**b**) Fully fabricated and assembled LaserPack containers on a CD before they dispense fluid. (**c**) Open LaserPacks with laser showing fluid dispensed on the CD.

**Figure 10 sensors-22-08955-f010:**
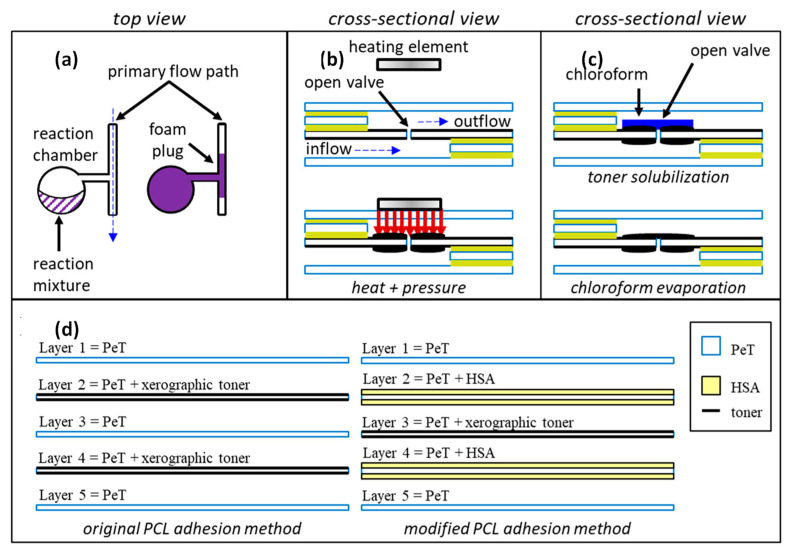
Schematic diagrams of the three types of microvalves described in [48]. Valve of type (**a**): Foam producing reagents are added to upstream reagent chambers within the device. Centrifugal pumping combines and mixes the two components. Upon mixing, the rapidly expanding polyurethane foam fills and blocks the target downstream channel. Valve of type (**b**): Heat and pressure are applied directly to a previously opened laser hole. Controlled compression and melting of the polymeric layers and adhesives form a permanent seal upon cooling. Valve of type (**c**): Chloroform is added to a previously opened laser patch, dissolving a portion of the xerographic toner. Rapid evaporation of the chloroform redeposits the toner into a uniform layer that covers the previously ablated laser hole. (**d**) Microdevices featured in this work consisted of five laminated PeT films that were prepared and assembled using the “print-cut-laminate” (PCL) method of fabrication. Cited from [47]. Permission was granted by the publisher.

**Figure 11 sensors-22-08955-f011:**
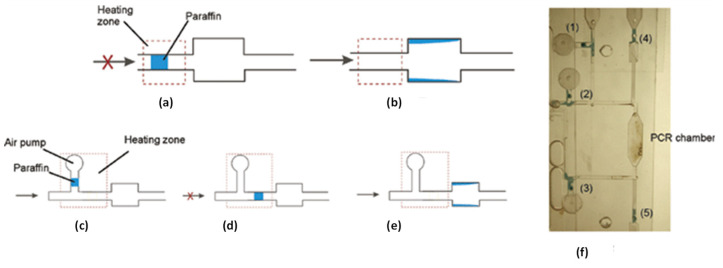
Schematic illustrations of a close–open paraffin microvalve design (**a**,**b**) and an open–close–open microvalve design (**c**–**e**). The former has a block of paraffin that initially closes the channel (**a**). To open the channel, the paraffin is melted using the heater underneath and moved downstream by the pressure from the upstream channel. Once the molten paraffin moves out of the heating zone, it starts to solidify on the wall of a wide channel section resulting in an open channel (**b**). The latter is a normally open valve with a block of paraffin connected to an air pocket that acts as a thermally actuated air pump (**c**). When the heater is activated, the air in the pocket expands and pushes the molten paraffin into the regulated channel. If the heater is turned off immediately, the paraffin solidifies in the main flow channel, resulting in a closed channel (**d**). The channel can be reopened by reactivating the heater (**e**). A photograph (**f**) of the PCR chamber surrounded by five paraffin-based microvalves:  valves 1–3 are open–close valves, and valves 4 and 5 are close–open valves. All valves are in “closed” position prior to initiating PCR. The PCB that provides thermal actuation to the valves is not shown here. Purified *E. coli*–bead complexes (brown) are retained in the PCR chamber. Permission granted by the publisher [62].

**Figure 12 sensors-22-08955-f012:**
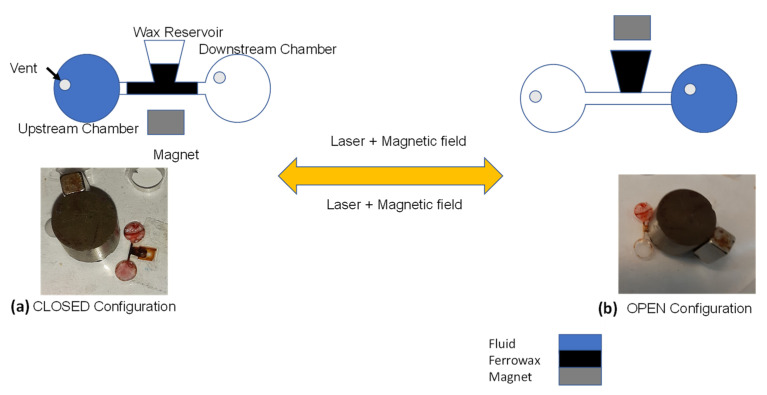
Magnet-based ferrowax microvalve. (**a**) the magnet near the valving channel, which attracts the molten ferrowax (melted with laser), creating a closed configuration of the valve. (**b**) Magnet near the wax reservoir, which collects the molten wax in the reservoir opening the valve to open configuration of the valve. The figure was reconstructed from a source cited [48].

**Figure 13 sensors-22-08955-f013:**
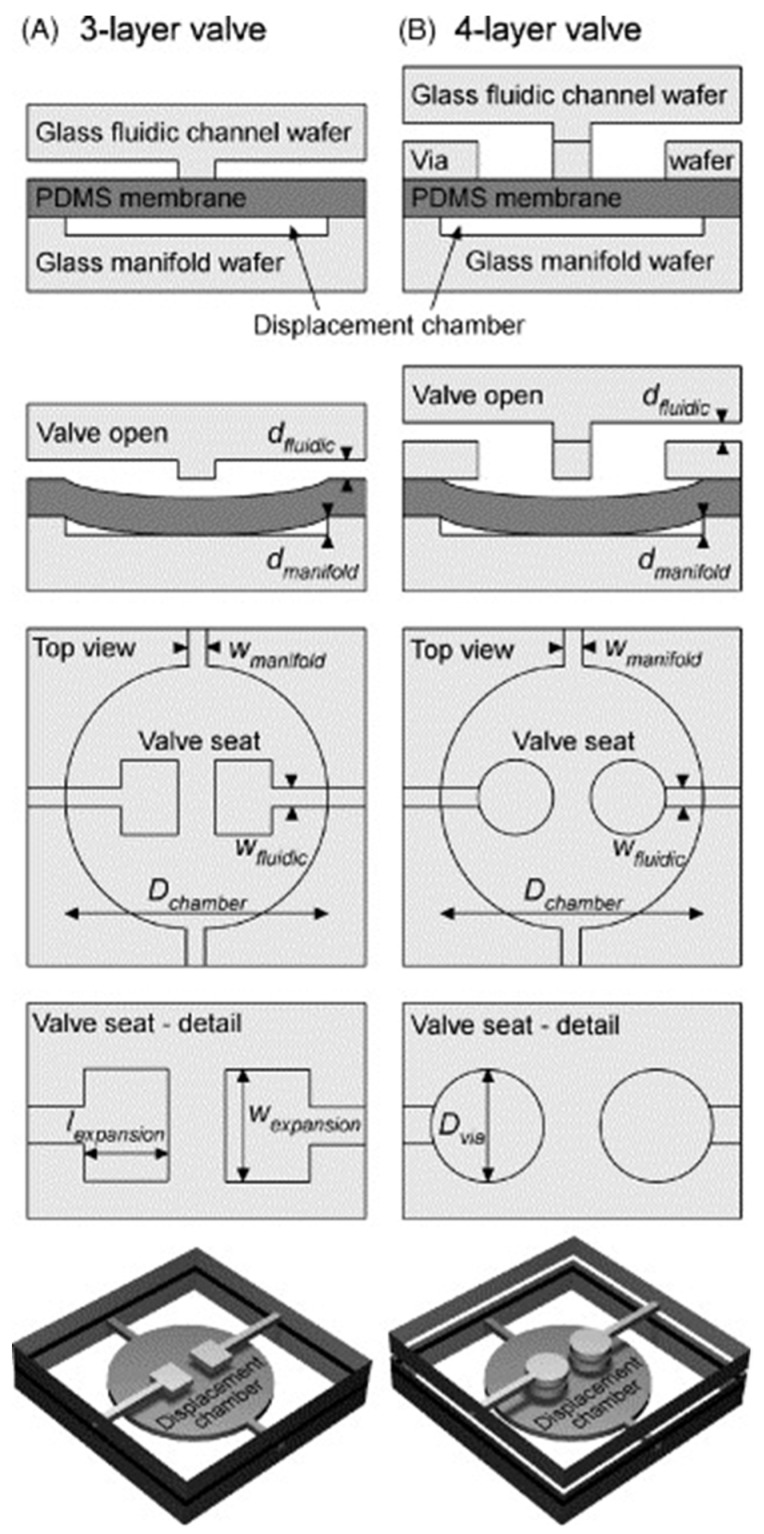
Cross-sectional, top, and oblique views of three-layer (**A**) and four-layer (**B**) monolithic PDMS membrane valves. Each valve consists of a glass manifold with an etched displacement chamber for pneumatic actuation, a working PDMS membrane, and a glass fluidic channel wafer containing the channel to be valved. In the three-layer topology, PDMS defines one surface of the valved channel. In the four-layer structure, the addition of the drilled via wafer defines all-glass fluidic channels with minimal fluid-PMDS contact. Permission granted by the publisher [52].

**Figure 14 sensors-22-08955-f014:**
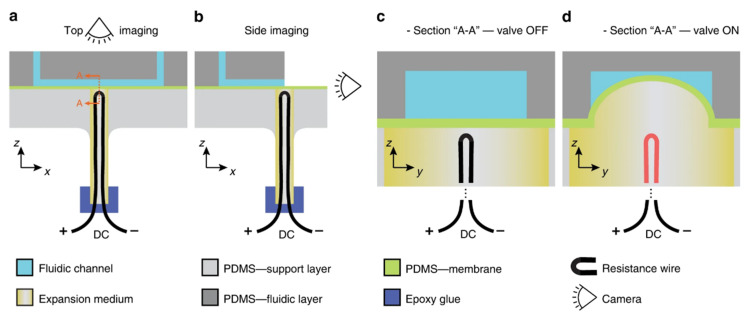
(**a**) The three-layer PDMS setup has a fluidic layer that houses the fluidic access ports and microchannels on the top. The middle layer houses the PDMS membrane that is ~200 μm thick; the bottom layer has a punched-hole with a glass capillary fitted. The capillary is filled with the expansion medium, and a resistance wire is inserted for heating. The capillary is sealed with epoxy glue. Imaging is from the top as shown. (**b**) The side imaging option is shown. During side imaging, the fluidic layer is cut in the middle for visualizing membrane deformation in real time. (**c**) Zoomed in A-A section view where the valve is in the OFF-state, no heating. (**d**) Zoomed in A-A section view where the valve is in the ON-state. Thermal expansion of the medium causes the membrane to block the flow in the fluidic layer. Permission granted by the publisher [59].

**Figure 15 sensors-22-08955-f015:**
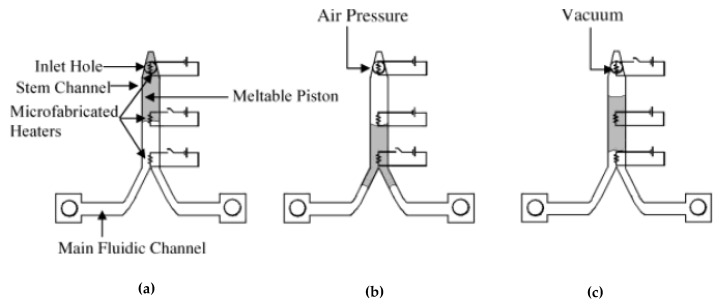
Schematic of the phase change valve operation. (**a**) Loading wax by actuating inlet port heater. (**b**) Closing valve by actuating the inlet port and stem channel heaters with pressure at inlet port. (**c**) Opening valve by actuating the stem channel and intersection heaters with vacuum at the inlet port. Permission granted by the publisher [63].

**Figure 16 sensors-22-08955-f016:**
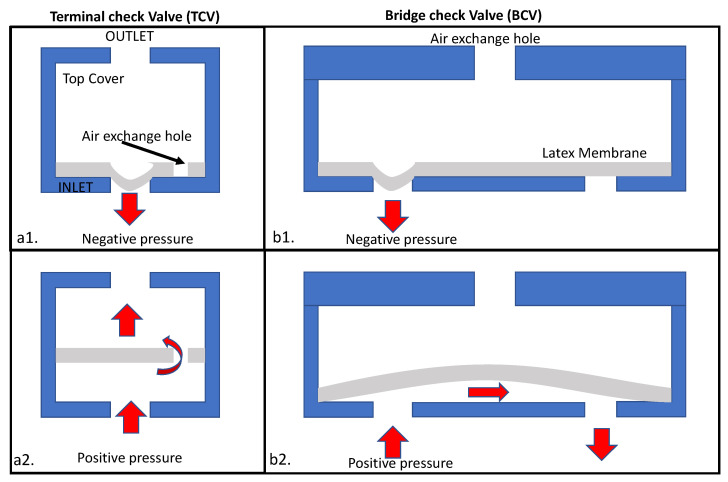
(**a1**) The first design of an air pressure-based check valve or the terminal check valve (TCV). Negative pressure is blocked by bending the latex membrane and blocking the flow. (**a2**) The TCV under positive pressure, which allows the fluid to pass through the air-exchange hole. This will occur only until the membrane is held in the middle. When it touches the outlet wall, the flow is blocked again. (**b1**) The bridge check valve or BCV. Negative pressure deforms the latex membrane and blocks the fluid flow. (**b2**) Positive pressure deforms the latex membrane to allow the fluid to pass into the outlet on the same side of the chamber. The figure was reconstructed from a source cited [64]. Permission was granted by the publisher to reproduce.

**Figure 17 sensors-22-08955-f017:**
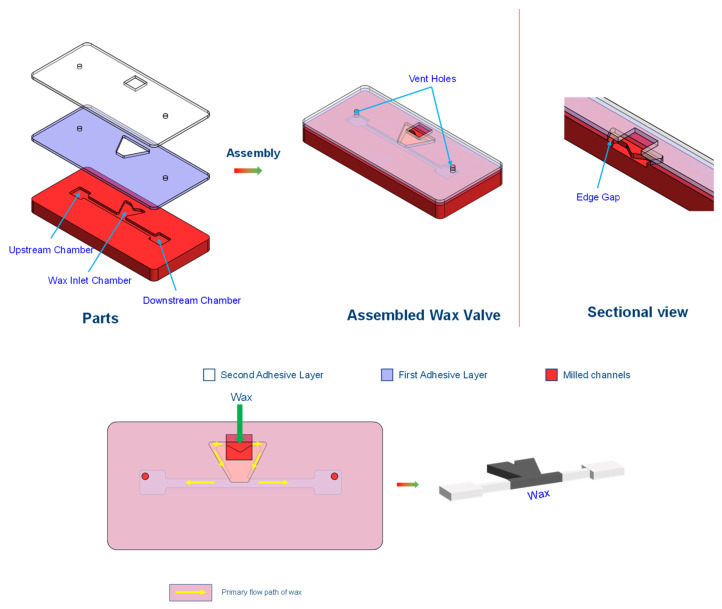
Assembly of the capillary-driven wax valve. The sectional view: location of the edge gaps and the main reservoir channel where the molten wax flows due to capillary action. Bottom figure: operation of the wax valve and how the wax flows into the valving channel. Vent holes are required in both upstream and downstream chambers. The figure was reconstructed from sources cited [48].

**Figure 18 sensors-22-08955-f018:**
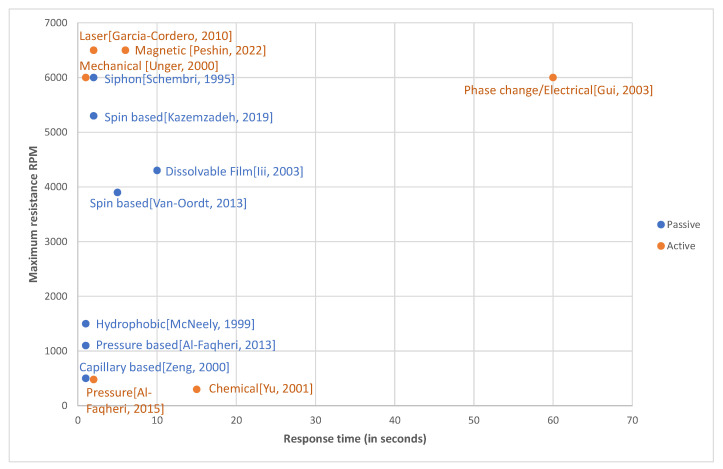
The various types of active and passive valves plotted on the axis of response time (in seconds) versus the maximum CD spin rate (in RPM) that the valves are able to withstand before bursting. The location of the valve to calculate the resistance RPM is 20 mm from the center of a disc that had a diameter of 80 mm [11,22,29,34,37,38,42,45,48,49,60,64,72].

## Data Availability

Not applicable.

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
