# Peer review of "Microvalves for Applications in Centrifugal Microfluidics"

_sensors, 2022, doi:10.3390/s22228955_

Round 1

Reviewer 1 Report

This review focuses on the various available fluidic valving systems applied in CD fluidic platforms. Those valving systems were categorized into either “active”, “passive”, or “hybrid” - based on their actuation mechanisms. Key topics of the valving systems such as actuation mechanism and governing physics were discussed and valving performance such as necessary disc spin rate for valve actuation, valve response time, and other parameters was compared. The applicability of some types of valves for specialized functions like reagent storage, flow control, and other applications is summarized.

The topic of CD fluidic platforms and fluidic valving systems has been researched for years and this hasn't been comprehensively reviewed recently. The authors have interpreted and presented the relevant results correctly and done comprehensive literature research covering the recent advances on the discussed topic. Besides, the manuscript is properly organized and well-written. Therefore, I would recommend the manuscript be accepted as it is or after a minor revision. There are a few minor comments for your consideration:

1.       The classification of table 2 is simple listing different types of valves used in centrifuge microfluidics. In this case, the author may consider combining table 1 and table 2 to provide a more informatic table for readers of interest.

2.       In the figure captions, the author should cite the specific articles used in the figure instead of citing a list of articles.

3.       All figures should use the same numbering format. In figure 2b, the author should crop the original figures instead of covering the original numbers.

4.       For the classification of valves, authors may add subheadings based on the control principles so the readers can have a better understanding of different kinds of valves.

5.       Some schematics describe the working mechanism of the valves and some schematics also introduce the manufacturing methods. Can authors summarise common fabrication methods for those valves with a table or a figure?

6.       As a critical review, the “critical” content seems inadequate. It would be better to add comments regarding not only their strength but also their weaknesses to some research mentioned in the article, especially for the most recent, advanced ones. The critical attitude should be followed throughout the whole article so that readers can understand the advances and weaknesses of the chosen examples, get valuable ideas, and form their perspectives on “what else can we do?” during the whole reading rather than realizing these ideas only at the end of the article. Such practice would better demonstrate the purpose of choosing certain studies for illustration, and would also make the logic of each paragraph clearer.

Reviewer 2 Report

This is a comprehensive review, even though the topic may not guarantee a broad readership. Anyway, the reviewer endorses the publication. 

Author Response

Dear Reviewer, thank you for your time and consideration.

Reviewer 3 Report

This manuscript is an extensive and comprehensive review of centrifugal microfluidics. It looks at several aspects of this technology in depth and its applications. The review is well written and well structured to lead readers outside the specific field into the filed of centrifugal microfluidics. It covers the literature well. 

I support publication as is. 

Reviewer 4 Report

The authors discussed various types of Valves and their applications in Centrifugal Microfluidics in this paper. Though, the content of the paper is adequate, I believe, there is a need of major improvement. Therefore, I suggest to publish this paper after a major revision.

1. The abstract and the motivation of the paper is not clear. Authors mentioned the title of the paper as "Microvalves for .......Microfluidics". Authors mentioned only 'valves', not 'microvalves' in the abstract. I could not find the reason why microvalve is important. Authors mostly focused on Valves instead of Microvalves. 

2.Insuffiient references. I suggest authors to go through the recent literatures and cite them accordingly. For example, 'A review of Microvalves', J. Micromech. Microeng. 16 (2006) R13–R39 doi:10.1088/0960-1317/16/5/R01 .

3.Section 2 in this paper discusses various types of valves and their applications. This paper will attract more readers if authors can put one more column in the table to show the foused application area of the valves.

4.In section 2, the authors are suggested to use the schematic/diagram consistently (use the schematic/diagram for all the type of valves mentioned in 2.1.1-2.1.5, instead of using it selectively). 

Round 2

Reviewer 4 Report

All the corrections and modifications are satisfactory. I recommend pulishing the draft without any further modifications.